# CD206[+] M2-like macrophages regulate systemic glucose metabolism by inhibiting proliferation of adipocyte progenitors

Allah Nawaz[1], Aminuddin Aminuddin [1,2], Tomonobu Kado[1], Akiko Takikawa[1], Seiji Yamamoto[3], Koichi Tsuneyama[4,5], Yoshiko Igarashi[6], Masashi Ikutani[7], Yasuhiro Nishida[1], Yoshinori Nagai[8,9], Kiyoshi Takatsu[8,10], Johji Imura[4], Masakiyo Sasahara[3], Yukiko Okazaki[11], Kohjiro Ueki[11,12], Tadashi Okamura[13,14], Kumpei Tokuyama[15], Akira Ando[15], Michihiro Matsumoto[16], Hisashi Mori[17], Takashi Nakagawa[18], Norihiko Kobayashi[19], Kumiko Saeki[19], Isao Usui[1], Shiho Fujisaka[1] & Kazuyuki Tobe[1]

Adipose tissue resident macrophages have important roles in the maintenance of tissue homeostasis and regulate insulin sensitivity for example by secreting pro-inflammatory or anti-inflammatory cytokines. Here, we show that M2-like macrophages in adipose tissue regulate systemic glucose homeostasis by inhibiting adipocyte progenitor proliferation via the CD206/TGFβ signaling pathway. We show that adipose tissue CD206[+] cells are primarily M2-like macrophages, and ablation of CD206[+] M2-like macrophages improves systemic insulin sensitivity, which was associated with an increased number of smaller adipocytes. Mice genetically engineered to have reduced numbers of CD206[+] M2-like macrophages show a down-regulation of TGFβ signaling in adipose tissue, together with up-regulated proliferation and differentiation of adipocyte progenitors. Our findings indicate that CD206[+] M2-like macrophages in adipose tissues create a microenvironment that inhibits growth and differentiation of adipocyte progenitors and, thereby, control adiposity and systemic insulin sensitivity.

[1] First Department of Internal Medicine, University of Toyama, 2630 Sugitani, Toyama-shi, Toyama 930-0194, Japan. [2] Department of Nutrition, Faculty of Medicine, University of Hasanuddin, Makassar, Kota Makassar, Sulawesi Selatan 90245, Indonesia. [3] Department of Pathology, University of Toyama, 2630 Sugitani, Toyama-shi, Toyama 930-0194, Japan. [4] Department of Diagnostic Pathology, University of Toyama, 2630 Sugitani, Toyama-shi, Toyama 930-0194, Japan. [5] Department of Pathology and Laboratory Medicine, Institute of Biomedical Sciences, Tokushima University Graduate School, 3-18-15 Kuramoto, Tokushima 770-8503, Japan. [6] Division of Kampo Diagnostics, Institute of Natural Medicine, University of Toyama, 2630 Sugitani, Toyama-shi, Toyama 930-0194, Japan. [7] Department of Immune Regulation, Research Center for Hepatitis and Immunology, Research Institute, National Center for Global Health and Medicine, 1-7-1 Kohnodai, Ichikawa, Chiba 272-8516, Japan. [8] Department of Immunobiology and Pharmacological Genetics, Graduate School of Medicine and Pharmaceutical Science for Research, University of Toyama, 2630 Sugitani, Toyama-shi, Toyama 930-0194, Japan. [9] JST, PRESTO, 4-1-8 Honcho, Kawaguchi, Saitama 332-0012, Japan. [10] Toyama Prefectural Institute for Pharmaceutical Research, 17-1 Nakataikouyama, Imiz-shi, Toyama 939-0363, Japan. [11] Department of Diabetes and Metabolic Diseases, Graduate School of Medicine, The University of Tokyo, 7-3-1 Hongo, Bunkyo-Ku, Tokyo 113-8655, Japan. [12] Department of Molecular Diabetic Medicine, Diabetes Research Center, National Center for Global Health and Medicine, 1-21-1 Toyama Shinjuku-ku, Tokyo 162-8655, Japan. [13] Department of Laboratory Animal Medicine, Research Institute, National Center for Global Health and Medicine, 1-21-1 Toyama, Shinjuku-ku, Tokyo 162-8655, Japan. [14] Section of Animal Models, Department of Infectious Diseases, Research Institute, National Center for Global Health and Medicine, 1-21-1 Toyama Shinjuku-ku, Tokyo 162-8655, Japan. [15] Doctoral Program in Sports Medicine, Graduate School of Comprehensive Human Sciences, University of Tsukuba, Tsukuba, Tennodai 1-1-1, Tsukuba, Ibaraki 305-8574, Japan. [16] Department of Molecular Metabolic Regulation, Diabetes Research Center, Research Institute, National Center for Global Health and Medicine, 1-21-1 Toyama, Shinjuku-ku, Tokyo 162-8655, Japan. [17] Department of Molecular Neuroscience, University of Toyama, 2630 Sugitani, Toyama-shi, Toyama 930-0194, Japan. [18] Department of Metabolism and Nutrition, University of Toyama, 2630 Sugitani, Toyama-shi, Toyama 930-0194, Japan. [19] Department of Disease Control, Research Institute, National Center for Global Health and Medicine, 1-21-1 Toyama, Shinjuku-ku, Tokyo 162-8655, Japan. Allah Nawaz & Aminuddin Aminuddin contributed equally to this work. Correspondence and requests for materials should be addressed to S.F. (email: shihof@med.u-toyama.ac.jp) or to K.T. (email: tobe@med.u-toyama.ac.jp)

White adipose tissue (WAT) markedly adapts to nutrient excess through adipocyte hypertrophy and hyperplasia[1–3]. The WAT expansion greatly affects the pathogenesis of obesity through different cellular mechanisms[4]. Adipocyte size is inversely related to insulin resistance[5], whereas the number of adipocytes is related to the pool size of adipocyte progenitors (APs). However, the cellular and molecular mechanisms regulating adipocyte size and number in vivo are largely unknown. Several groups, including our laboratory, have reported that M1-like inflammatory macrophages regulate the expression of angiogenic genes in preadipocytes[3, 6], suggesting interactions between macrophages and APs. It is still unknown how the proliferation and differentiation of APs are regulated by M2-like macrophages within WAT, thus controlling the insulin sensitivity.

Obesity is associated with a phenotypic transformation of macrophages, from anti-inflammatory M2 to pro-inflammatory M1 macrophages, thereby causing insulin resistance[1, 7, 8]. M2 macrophages are required for maintenance of homeostasis, tissue remodeling, and metabolic adaptation under nutrient surplus conditions[9, 10], but it is largely unknown how macrophages participate in progenitor activation and adipogenesis.

TGFβ and related factors control the development, growth and function of diverse cell types. TGFβ is often secreted by niche cells, thereby inducing hibernation of tissue stem cells such as hematopoietic and melanocyte stem cells[11, 12]. WAT-derived TGFβ1 reportedly contributes to insulin sensitivity, while blockade of TGFβ/smad 3 signaling induces browning to protect against obesity and diabetes[13]. Adipose tissues of obese mice and humans showed higher TGFβ1 expression[14–16]. We hypothesized that M2-like macrophages might be involved in the regulation of remodeling of WAT via TGFβ signaling.

In the current study, we have successfully performed partial but specific depletion of CD206[+] M2-like macrophages without affecting either the number or functions of M1 macrophages, and without affecting body weights or overall adiposity. We show that CD206[+] M2-like macrophages have pivotal roles in WAT remodeling by modulating APs proliferation and differentiation into adipocytes through TGFβ signaling, providing a niche for APs. We further determin the specific involvement of CD206[+] M2-like macrophages in terms of insulin sensitivity and adipose tissue remodeling both under normal chow (NC) and high-fat diet (HFD)-fed conditions. Thus, CD206/TGFβ signaling is pivotal players in modulating APs proliferation and differentiation to adjust adiposity and systemic insulin sensitivity.

## Results

**CD206 is a specific marker for M2-like ATMs**. To investigate the involvement of M2-like ATMs in the regulation of adipose tissue dynamics during metabolism-associated remodeling/repairing, we looked for a specific marker for M2-like ATMs. We have previously shown that the vast majority of ATMs are CD206[+] M2-like macrophages, but the ratio of CD206[+] M2-like macrophages in F4/80-positive macrophage and F4/80-negative non-macrophage populations was not evaluated. To address these issues, we collected stromal vascular fractions (SVF) populations from epididymal WAT (eWAT) and subjected them to flow cytometric analysis. Cells were gated on CD45-positive cells and expression of CD206 and F4/80 on these cells were analyzed. Flow cytometry analysis showed that the almost all CD206-positive populations are F4/80-positive (Fig. 1a and Supplementary Fig. 1), indicating that CD206[+] cells in adipose tissues are macrophages, but not cells of other lineages. Consistently, F4/80 messengerRNA (mRNA) expression levels in F4/80[+]CD206[+] populations compared with those in total SVF populations (Fig. 1b, black/blue ratios) were equivalent to the relative levels of

the well-characterized M2-like macrophage markers CD163, and MgL2. We determined that the F4/80[+]CD206[−] population expressed higher level of CD11c, TNFα, IL-6, Zbtb46 mRNA (Fig. 1b, red) than the F4/80[+]CD206[+] population, indicating that the former includes substantial amounts of M1-like macrophages and dendritic cells. In any event, CD206[+] populations, which comprise the major population in ATMs, are exclusively M2-like macrophages and not M1-like macrophages or non-macrophage populations. Thus, CD206 provides an ideal marker to target M2-like macrophages in adipose tissue.

**CD206[+] M2-like ATMs depletion promote adipose tissue metabolism**. We recently reported that CD206[+] M2-like macrophages were selectively depleted in CD206DTR mice, in which a human diphtheria toxin receptor (DTR) expression unit was knocked in at the CD206 promoter locus[17] (Fig. 1c left, Supplementary Fig. 2a–d). The CD206DTR and wild-type (WT) mice showed similar body weights during the observation period from 6–14 weeks of age in the absence of diphtheria toxin (DT) (Supplementary Fig. 1e).

We administered DT three times every other day and found that injection of DT with dose 3ng/gBW (Fig. 1c, right) specifically reduces the number of CD206[+] M2-like macrophages without affecting body weight (Fig. 1d), the weight of eWAT or inguinal WAT (iWAT) (Fig. 1e) or even food intake (Fig. 1f). DT administration did not affect the number of M1-like macrophages and expression levels of M1-like markers in the eWAT (Fig. 1g and h and Supplementary Fig. 3), although minor alterations in the expression of natural killer cells and eosinophils were observed in the eWAT of CD206[+] M2-like macrophages-reduced mice (Supplementary Fig. 4a). In addition, we found that the expression of fibrosis related marker genes such as Col1a1 and Acta2, were downregulated in CD206[+] M2-like macrophages-reduced mice (Supplementary Fig. 4b). However, immunohistochemical analysis revealed that CD206[+] M2-like amcrophages depletion did not alter adipose tissue fibrosis (Supplementary Fig. 4c). The expression of CD206 and other M2-like macrophage markers were also downregualted in iWAT of DT-treated CD206DTR mice (Supplementary Fig. 4d). Decline of CD206[+] M2-like macrophages were also observed in bone marrow (BM), the liver and skeletal muscle of CD206[+] M2-like macrophages-reduced mice (Supplementary Fig. 4e–g). Flow cytometric analysis of the peritoneal cavity macrophages revealed that CD206[+] M2-like macrophages were also depleted (Supplementary Fig. 5a). In addition, gene expression and flow cytometric analysis of BM shows that the number of eosinophils, natural killers cells, and granulocytes was unaffected (Supplementary Fig. 5b–d). Thus, the current protocol provides an effective approach for systemic reduction of CD206[+] M2-like macrophages without affecting the numbers of other lineage cells, body weight, adiposity, or food intake (Fig. 1d–h).

We evaluated the physiology of the adipose tissues by examining the size and number of adipocytes. In CD206[+] M2-like macrophages-reduced mice, the size of adipocytes was significantly reduced (Fig. 2a), while the numbers of adipocytes (Fig. 2b) and SVF populations (Fig. 2c) were increased in eWAT. Flow cytometric analysis revealed that BrdU-uptake was increased in the CD45[−] SVF cells of CD206[+] M2-like macrophages-reduced mice (Fig. 2d and Supplementary Fig. 5d), suggesting that the depletion of CD206[+] M2-like macrophages resulted in the proliferation of CD45[−] cells. In agreement with this, expression of cell cycle indicators, such as cyclins and Ki-67, was also increased in the eWAT of CD206[+] M2-like macrophages-reduced mice (Fig. 2e). We also found an increased number of Ki-67- and BrdU-positive cells in CD206[+] M2-like

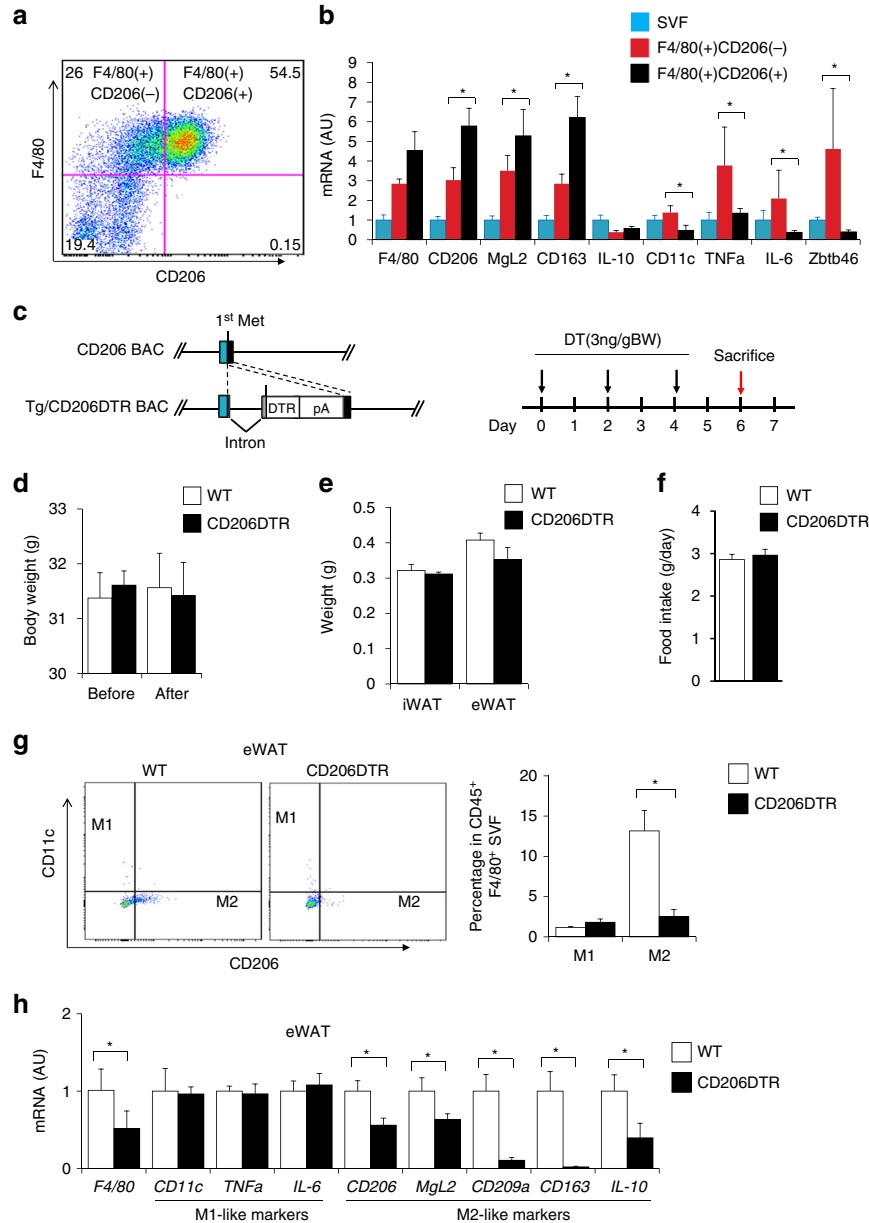

**Fig. 1** Characterization of CD206+ ATMs. **a** Representative flow cytometry analysis of F4/80 and CD206 expression in the SVF of eWAT from WT mice (n = 4–5). Full gating strategy is given in Supplementary Fig. 1. **b** Relative mRNA expression of SVF and FACS-sorted F4/80+CD206+ from eWAT of WT mice. Sorted cell populations based on the expression of CD206 and F4/80 on the SVF CD45+ cells from eWAT of NC mice (n = 4–5 mice per group). The data are shown as the means ± SEM. *P < 0.05 compared by ANOVA. **c** Schematic representation of CD206 and Tg/CD206DTR inserted transgene in BAC DNA (Tg, transgenic). Exons and the coding region of CD206 are indicated by open box and black boxes, respectively. Met is the initiation site of translation in CD206 and DTR. The inserted intron, DTR and polyA signals (pA) are indicated (*left panel*). A schematic diagram of the DT injection method for metabolic analysis of CD206DTR mice (*right panel*). **d** Body weights before and after DT injections, and **e** weight of adipose tissue and **f** food intake after DT injection period (n = 4–6 mice per group). **g** Representative flow cytometry analysis of CD11c and CD206 expression in F4/80+ SVF cells of eWAT from WT and CD206DTR mice (*left panel*). Percentages of M1 and M2 macrophages were calculated (*right panel*) (n = 5–6 mice per group). Full gating strategy is given in Supplementary Fig. 3. **h** The relative mRNA expression of M1/M2 macrophage markers in the eWAT after DT treatment (n = 5–6 mice per group). The data are shown as the means ± SEM. *P < 0.05, **P < 0.01 compared with littermates by Student's t-test

macrophages-reduced mice (Fig. 2f). Taken together, these data suggested CD206+ M2-like macrophages depletion triggered the proliferation of CD45− SVF.

**CD206+ M2-like ATMs regulate adipocyte progenitor's proliferation**. We examined which type of CD45− cells in the eWAT were proliferating after CD206+ M2-like macrophages depletion. As macrophages are reportedly involved in the regulation of

progenitor activity or stem cell niche activity[3, 6, 18–20], we investigated the possible involvement of CD206+ M2-like macrophages in the control of APs proliferation. As shown in Fig. 3a, expression of a series of AP markers including *PDGFRa*, *CD24*, *Sca-1*, and *Pref-1*[21–24], and mesenchymal stem cell (MSC) markers including *CD105* and *CD90*[25, 26] were up-regulated in eWAT of the CD206+ M2-like macrophages-reduced mice. These data indicated that proliferating cells might be APs. To directly assess the proliferation state of APs, the CD45−CD31−Sca-1+

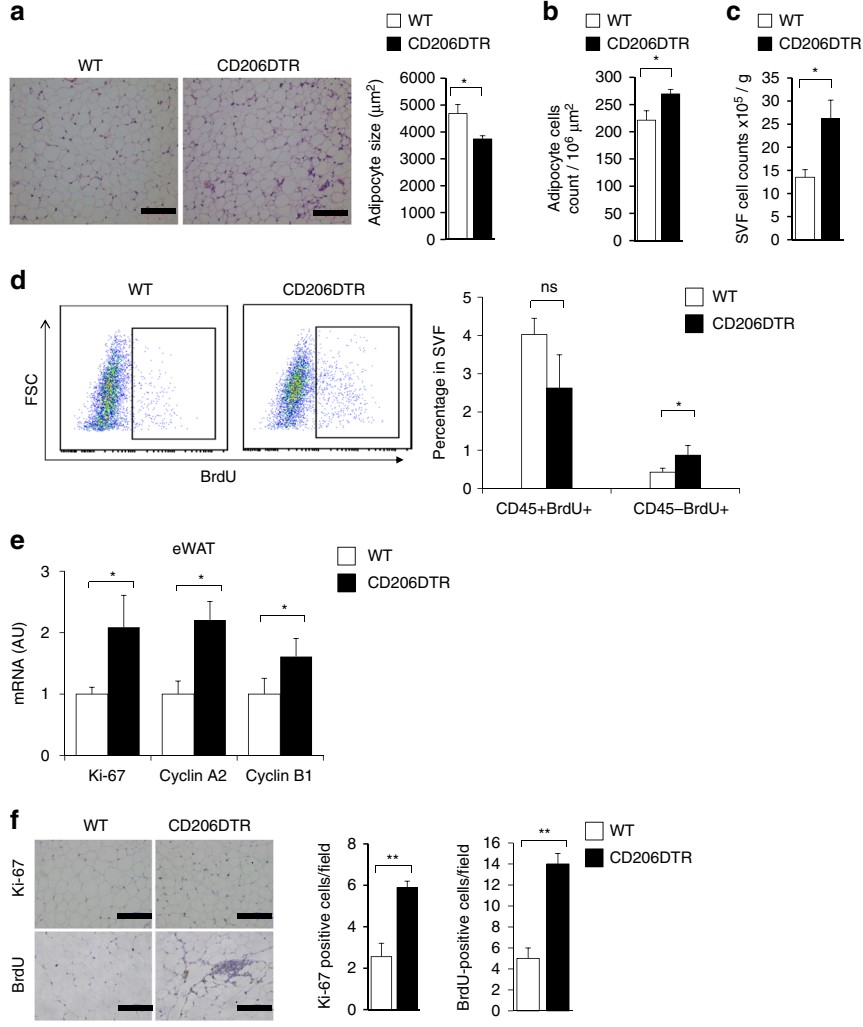

**Fig. 2** Depletion of CD206[+] ATMs induces generation of smaller adipocytes. **a** Representative sections of eWAT from WT and CD206DTR were stained with hematoxylin and eosin (HE) (*left panel*) and adipocyte size (*right panel*); Scale bar, 100 μm. **b** Cell count measurements in the eWAT. **c** Cell count of SVF obtained from eWAT of DT-treated WT and CD206DTR mice ($n = 4$–5 mice per group). **d** Representative flow cytometry analysis of BrdU[+] cells in CD45[−] SVF from WT and CD206DTR mice (*left panel*). Percentages of CD45[+]/BrdU[+] and CD45[−]/BrdU[+] fractions in SVF from WT and CD206DTR mice were calculated (*right panel*) ($n = 4$–5 mice per group). Full gating strategy is given in Supplementary Fig. 5e. **e** Relative mRNA expression of cell cycle and proliferating marker genes in the eWAT ($n = 4$–7 per group). **f** Representative paraffin eWAT sections of Ki-67 immunostaining counterstained with eosin (*bluish*) (*upper panel*), and BrdU uptake (*lower panel*). Scale bar, 100 μm. Quantification (right panel; $n = 3$–6 per group). The data are shown as the means ± SEM. *$P < 0.05$, **$P < 0.01$, ns = non-significant compared with littermates by Student's t-test

non-endothelial fraction was purified using magnetic activated cell sorting (MACS) (Supplementary Fig. 6a). Sca-1[+] fractions showed even larger increments in the expression of cell cycling indicator genes under CD206[+] M2-like macrophages reduction than did Sca-1[−] fractions (Fig. 3b), indicating that the proliferation state of APs was indeed activated in CD206[+] M2-like macrophages-reduced mice. In agreement with this, flow cytometry analysis revealed that the number of Sca-1/PDGFRα double positive populations[23, 27] was increased (Fig. 3c and Supplementary Fig. 6b) and cyclin gene expression levels in APs fraction (Fig. 3d) were up-regulated. For further confirmation, the fate of proliferating cells was studied by injecting the mice with EdU; this was specifically incorporated into S-phase cells, which were traced over time. After 2 h of EdU injection, EdU[+] cells and PDGFRα[+] cells were detected with higher frequency in CD206[+] M2-like macrophages-reduced mice than in WT mice (Fig. 3e). After 96 h following injection, EdU[+] nuclei were detected in cells that were positive for perilipin, a lipid droplet-coating protein (Fig. 3f). Thus, proliferating APs in CD206[+] M2-

like macrophages-reduced mice were indeed differentiated into mature perilipin-positive adipocytes. When stained by CellMask™ Green Plasma Membrane Stain, we found that some EdU[+] nuclei were not separated from the lipid droplets by a plasma membrane (Fig. 3g), a characteristic of terminally maturated adipocytes with high lipid-storing capacities[28]. Thus, CD206[+] M2-like macrophages reduction promotes the proliferation of APs, which are subsequently differentiated into mature adipocytes.

**TGFβ signaling in CD206[+] M2-like ATMs-based APs proliferation.** M2 macrophages express fairly large amounts of TGFβ[29–31] and TGFβ is reportedly involved in inhibiting proliferation/differentiation of various cell types including preadipocytes, melanocyte- and hematopoietic-stem cells[11, 12, 32]. Enhanced TGFβ signaling is also reportedly associated with obesity[13–16, 33]. We therefore examined the possible involvement of TGFβ signaling in the up-regulated APs proliferation in

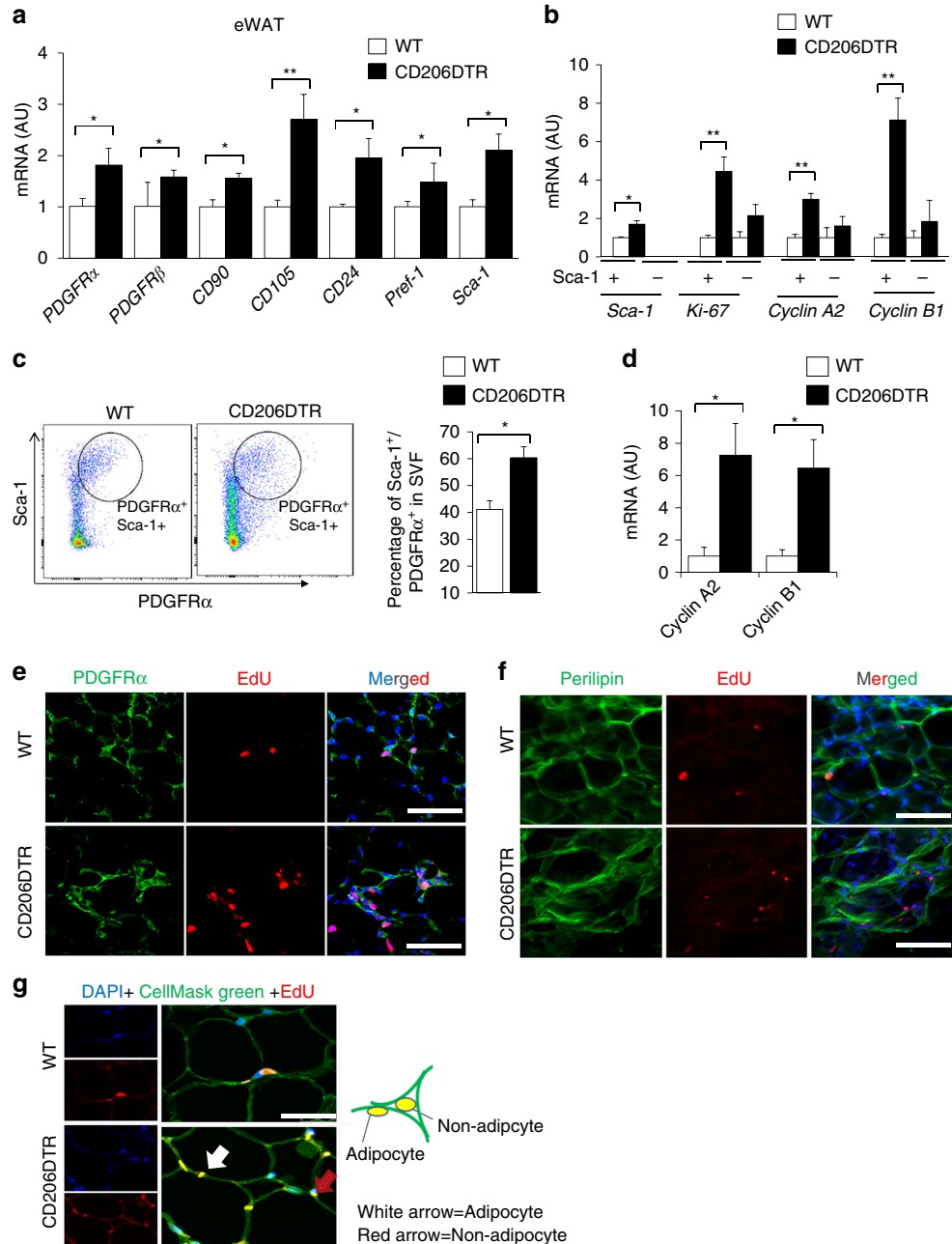

**Fig. 3** Effects of CD206+ M2-like macrophages depletion on cell proliferation. **a** Relative mRNA expression of mesenchymal stem cell (MSC) and AP markers in the eWAT ($n = 4$–6 mice per group). **b** Relative mRNA expression of MACS purified Sca-1 positive fraction ($n = 3$ mice per group). **c** Representative flow cytometry analysis of Sca-1/PDGFRα double positive cells in the SVF of eWAT from DT-treated WT and CD206DTR mice. Percentage of Sca-1/PDGFRα double positive fraction in the SVF of these mice was calculated (right panel; $n = 3$–5 per group). Full gating strategy is given in Supplementary Fig. 6b. **d** Relative mRNA expression of flow cytometry sorted Sca-1/PDGFRα double positive fraction in the eWAT ($n = 3$–4 mice per group). **e** Representative confocal images of eWAT frozen sections triple-stained for PDGFRα, EdU and DAPI 2 h after EdU injection. The nuclei were counterstained with DAPI (*blue*). *Scale bar*, 25 μm. **f** Representative images of frozen eWAT sections triple-stained for perilipin, EdU and DAPI 96 h after EdU injection. The nuclei were counterstained with DAPI (*blue*). The images were taken with Leica TCS-SP5 (Oil 63×). *Scale bar*, 25 μm. **g** High resolution confocal imaging of CellMask™ Green Plasma Membrane Staining of frozen eWAT sections. The plasma membrane was stained with Cell Mask Green (*green*), proliferating cells were stained with EdU (*red*) and the nuclei were stained with DAPI (*blue*). *White arrow* shows adipocytes and *red arrow* shows non-adipocytes. The images were taken with a Leica TCS-SP5 (Oil 63×). *Scale bar*, 25 μm. (*White arrows* = adipocyte, *Red arrow* = Non adipocytes). The data are shown as the means ± SEM. *$P < 0.05$, **$P < 0.01$ compared with littermates by Student's $t$-test

CD206+ M2-like macrophages-reduced mice. Among the genes involved in TGFβ signaling, TGFβ1 was abundantly expressed in CD206+ M2-like macrophages (Fig. 4a). Confocal imaging studies demonstrated co-localization of CD206+ M2-like ATMs with TGFβ immunostaining in eWAT (Fig. 4b). Furthermore, we also found that the number of cells that co-express CD206 and TGFβ1

was reduced in CD206+ M2-like macrophages-depleted mice (Fig. 4c). CD206+ M2-like macrophages were located in close proximity to PDGFRα+ APs, as shown in Pdgfrα-CreERT2-eGFP (PRa) mice and WT mice (Supplementary Fig. 6c, d), consistent with a previous report[3]. Co-localization of APs with p27Kip1, a downstream factor of TGFβ signaling for cell growth inhibition,

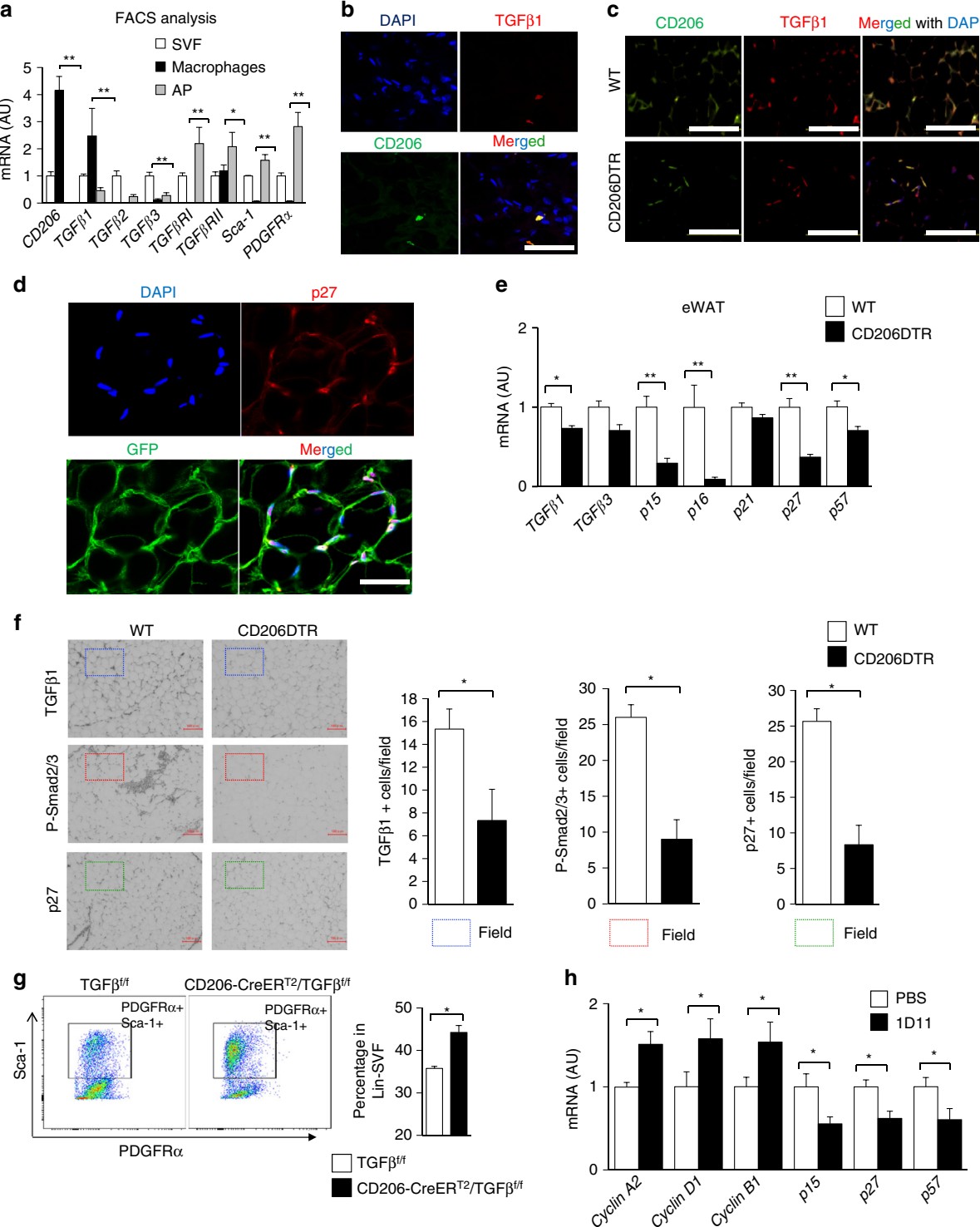

**Fig. 4** TGFβ signaling is reduced after CD206⁺ M2-like macrophages depletion. **a** Relative mRNA expression of *CD206, TGFβ1, TGFβRI*, and *TGFβRII*, in the SVF, macrophages (CD206⁺), and APs purified from eWAT of WT mice by using FACS (*n* = 4–5 per group). **b** High resolution confocal imaging of frozen eWAT sections from WT mice. *Scale bar*, 25 μm. **c** Representative images of eWAT paraffin sections stained for CD206 and TGFβ1 (*Scale bar*, 200 μm) in DT-treated CD206DTR and WT littermate. **d** High resolution confocal imaging of frozen eWAT sections from WT mice stained for GFP vs. p27. The images were taken with a Leica TCS-SP5 (Oil 63×). *Scale bar*, 25 μm. **e** Relative expression of TGFβ signaling genes in eWAT. **f** Representative immunohistochemical staining of the eWAT paraffin section for TGFβ1, phospho-smad2/3 (P-Smad2/3), and p27. *Scale bar* 100 μm. Quantification (*right panel*) (*n* = 3–4 per group). **g** Representative flow cytometry analysis of Sca-1/PDGFRα double positive population in the SVF of eWAT from CD206-CreER^T2^/TGFβ^flox/flox^ and TGFβ^flox/flox^ control mice. Percentage of Sca-1/PDGFRα double positive fraction was calculated (*right panel; n* = 4–5 per group). Full gating strategy is given in Supplementary Fig. 7d. **h** The relative mRNA expression of cell cycle and TGFβ signaling genes from eWAT samples collected after direct injection of a single dose of TGFβ neutralizing antibody (1D11) into the eWAT (*n* = 3). The data are shown as the means ± SEM. *\*P* < 0.05, *\*\*P* < 0.01 compared with littermates by Student's *t*-test

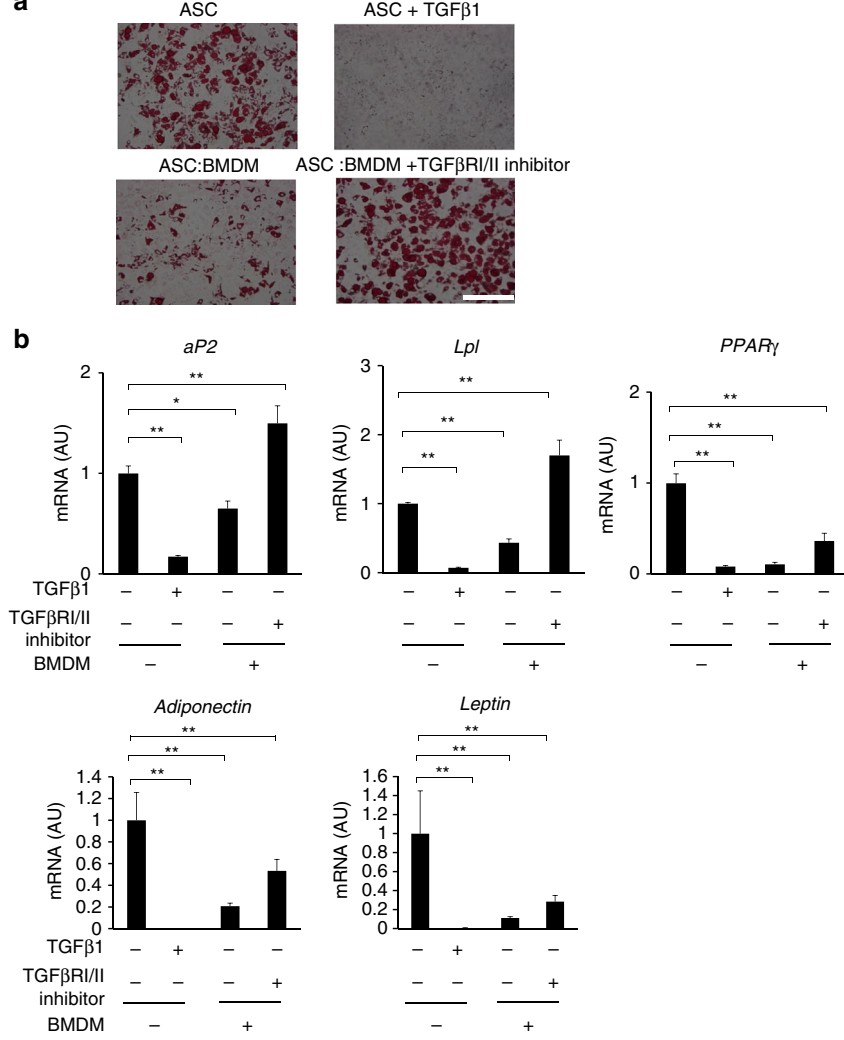

**Fig. 5** In vitro coculture of BMDM and ASC. **a** Oil *red* O staining of ASC and ASC:BMDM co-culture after adipogenesis stimulation for 12 days. Addition of TGFβ1 (*right upper panel*) and BMDM to ASC (*left lower panel*) inhibited adipogenesis. BMDM inhibitory effect on ASC adipogenesis was released by addition of TGFβRI/II inhibitor. *Scale bar*, 200 μm. **b** The relative expression levels of late adipogenesis markers. The data were calculated from experiments of 3–6 dishes per group. The data are shown as the means ± SEM. *$P < 0.05$, **$P < 0.01$ compared with control by ANOVA

was also confirmed in PRa mice (Fig. 4d). Expression of other TGFβ signaling downstream factors including p21, p15, p16, p27, and p57 were down-regulated in eWAT in CD206+ M2-like macrophages-reduced animals (Fig. 4e). In addition, phospho-smad2/3 (P-Smad2/3) and p27$^{Kip1}$ protein expression was reduced by CD206+ M2-like macrophages reduction in eWAT (Fig. 4f), suggesting that CD206+ M2-like macrophages suppress APs proliferation via the TGFβ signaling pathway. To validate this hypothesis, we generated a genetically engineered mouse in which TGFβ1 expression was specifically depleted in CD206+ M2-like macrophages (CD206−CreER$^{T2}$/TGFβ$^{flox/flox}$ mice) (Supplementary Fig. 7a). Administration of tamoxifen significantly reduced TGFβ1 expression by 20–30% in CD206+ M2-like macrophages in the CD206-CreER$^{T2}$/TGFβ$^{flox/flox}$ mice (Supplementary Fig. 7b); the expression of cell cycle-associated genes and APs marker genes (Supplementary Fig. 7c) and the number of Sca-1/PDGFRα double positive populations in eWAT were all up-regulated in these mice (Fig. 4g and Supplementary Fig. 7d). Immunostaining analysis shows that fibrosis of eWAT was decreased in tamoxifen treated CD206-CreER$^{T2}$/TGFβ1$^{flox/}$$^{flox}$ mice compared with tamoxifen treated TGFβ1$^{flox/flox}$ control mice (Supplementary Fig. 8). Moreover, the eWAT of WT mice

injected with an anti-TGFβ1,2,3 neutralizing monoclonal antibody showed increments in the expression of cell cycle-related genes, with reciprocal decrements in TGFβ downstream gene expression (Fig. 4h). To further confirm the impact of TGFβ signaling, we performed inhibitor analysis in vitro. Adipose tissue-derived stem cells (ASCs) collected from the iWAT of WT mice were co-cultured with CD206+TGFβ1+ M2-like macrophages, which were produced from bone marrow-derived macrophages (BMDM) by treating with IL-4 and PGE2. We also found that BMDM supplemented with IL-4 and PGE2 could be differentiated into M2-like macrophage, which highly expresses CD206 and TGFβ1 (Supplementary Fig. 9a–c). The presence of LY2109761 (a TGFβRI/II inhibitor) abrogated the inhibitory effect of BMDM on adipogenesis of ASCs (Fig. 5a, b) although LY2109761 per se, or anti-TGFβ1,2,3 enhanced PDGFRα+ cell proliferation (Fig. 6a, b). Thus, CD206/TGFβ signaling regulate the APs proliferation and subsequent adipogenesis.

**CD206+ M2-Like ATMs depletion improve glucose metabolism in lean mice.** We have previously showed that the majority of macrophages were M2-type in non-obese states, whereas the

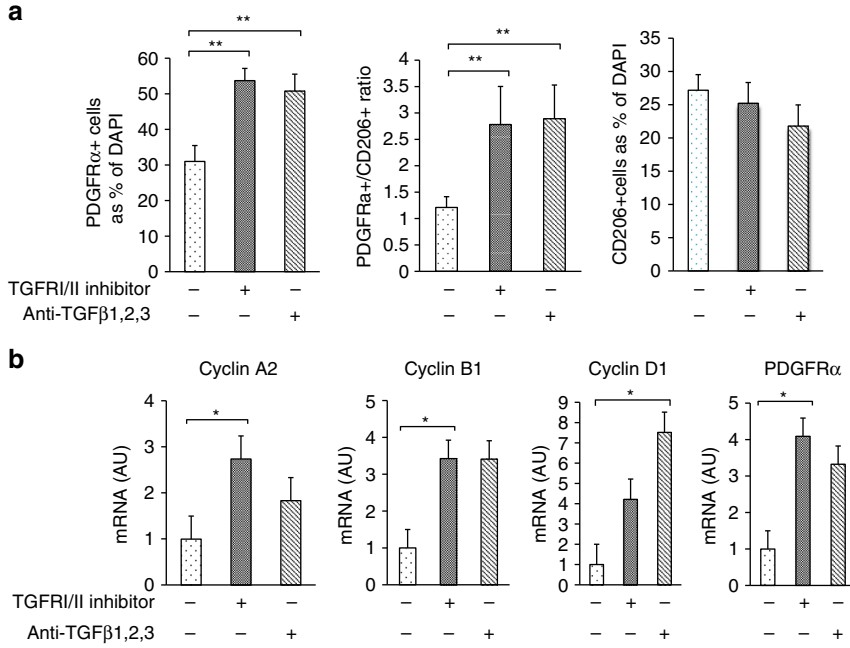

**Fig. 6** Reversal of BMDM-induced inhibition of adipogenesis. **a** Quantification of PDGFRα and CD206 in in vitro co-cultures of ASC and BMDM after the addition of a TGFβRI/II inhibitor (LY2109761) and anti- TGFβ1,2,3 antibody to the medium for 48 h. The PDGFRα and CD206+ macrophages were both counted as the percentages of DAPI-positive cells. **b** The relative mRNA expression levels of cell proliferation-related genes and *PDGFRα*. The data were calculated from experiments of 3–6 dishes per group. The data are shown as the means ± SEM. *$P < 0.05$, **$P < 0.01$ compared with control by ANOVA

majority of macrophages were M1 type in obese states[34]. As M2 ATMs are considered to be metabolically 'favorable' ATMs[35, 36] compared with M1 pro-inflammatory macrophages, it was expected that decline of CD206/TGFβ might impair glucose metabolism despite enhanced APs proliferation. Surprisingly, glucose tolerance and insulin sensitivity were improved in CD206+ M2-like macrophages-reduced mice (Fig. 7a–c). This unexpected finding was indeed a specific outcome of CD206+ M2-like macrophages reduction, as a glucose tolerance test without DT administration did not reveal any difference between WT and CD206DTR mice (Supplementary Fig. 10a). In CD206+ M2-like macrophages-reduced mice, the levels of Akt phosphorylation were up-regulated in eWAT, liver and skeletal muscle after insulin injection (Fig. 7d–f and Supplementary Fig. 10b). Moreover, expression levels of the genes associated with metabolically favorable states were also up-regulated in eWAT and skeletal muscle, while gluconeogenesis-related gene expressions were down-regulated in the liver (Fig. 7g–i). In addition, gene expression analysis revealed that expressions of markers for smaller adipocytes were consistently up-regulated in the eWAT, reflecting the increment in smaller adipocytes. Expression levels of adipogenesis-related transcription factors including *C/EBP-δ* and *C/EBP-α* were also markedly up-regulated in the eWAT (Fig. 7g). Thus, decline in CD206/TGFβ signaling promoted adipogenesis to provide an increment in smaller adipocytes in the WAT, thereby improving insulin sensitivity. In genetically engineered CD206-CreER[T2]/TGFβ[flox/flox] mice, the number of smaller adipocytes was also increased after tamoxifen treatment (Supplementary Fig. 10c), supporting our hypothesis about the involvement of CD206/TGFβ signaling in glucose metabolism. Thus, decline of CD206/TGFβ signaling improves glucose metabolism in lean mice.

**CD206+ M2-like ATMs depletion improve glucose metabolism in obese mice.** Next, we examined the impact of CD206+ M2-like macrophages reduction on glucose metabolism under HFD-fed conditions. WT and CD206DTR mice were subjected to HFD challenge for 16 weeks, with subsequent DT administration. Both WT and CD206DTR mice showed no difference in food intake (Supplementary Fig. 11a) or body weight gain (Supplementary Fig. 11b) during the 16 weeks of HFD-fed periods, but decreased weight gain was observed in CD206+ M2-like macrophages-reduced mice compared with WT after DT administration (Supplementary Fig. 11c). Notably, CD206+ M2-like macrophages-reduced mice showed improved glucose tolerance and insulin sensitivity even during HFD-fed periods (Fig. 8a and b). Consistent with this, they showed an increased number of smaller adipocytes with a reduced number of crown-like structures (CLS) compared with the WT control group (Fig. 8c–e). To determine the tissues responsible for amelioration of insulin resistance, we performed hyperinsulinemic-euglycemic clamp studies in DT-treated CD206DTR and WT mice (Fig. 8f). We determined that these mice showed significant increments in glucose infusion rate (GIR) and rate of whole-body glucose disappearance (Rd) than WT control mice without any significant suppressions in hepatic glucose production (HGP) (Fig. 8f), indicating that skeletal muscle/adipose tissues are responsible for metabolic improvement. Flow cytometric analysis further confirmed the reduced macrophage infiltration in response to HFD challenge (Fig. 9a and Supplementary Fig. 12a). The eWAT of CD206+ M2-like macrophages-reduced mice showed up-regulated expressions of adipogenesis-related transcription factor genes (e.g., *C/EBP-α, C/EBP-δ, PPARγ*), genes that are associated with metabolically favorable states (e.g., *PGC-1α, PGC-1β, Glut4*), APs and MSC marker genes (e.g., *Sca-1, PGDFRα, CD105*) and cell cycle-related genes (e.g., *Ki-67, Cyclin A2, Cyclin B1*), along with down-regulation of pro-inflammatory M1 macrophage markers (e.g., *CD11c, MCP1, NOS2, IL-6*, and *TNFα* (Fig. 9b–d). Thus, CD206+ M2-like macrophages play crucial roles in regulating glucose metabolism in HFD-fed obese mice, as well as NC-fed lean mice.

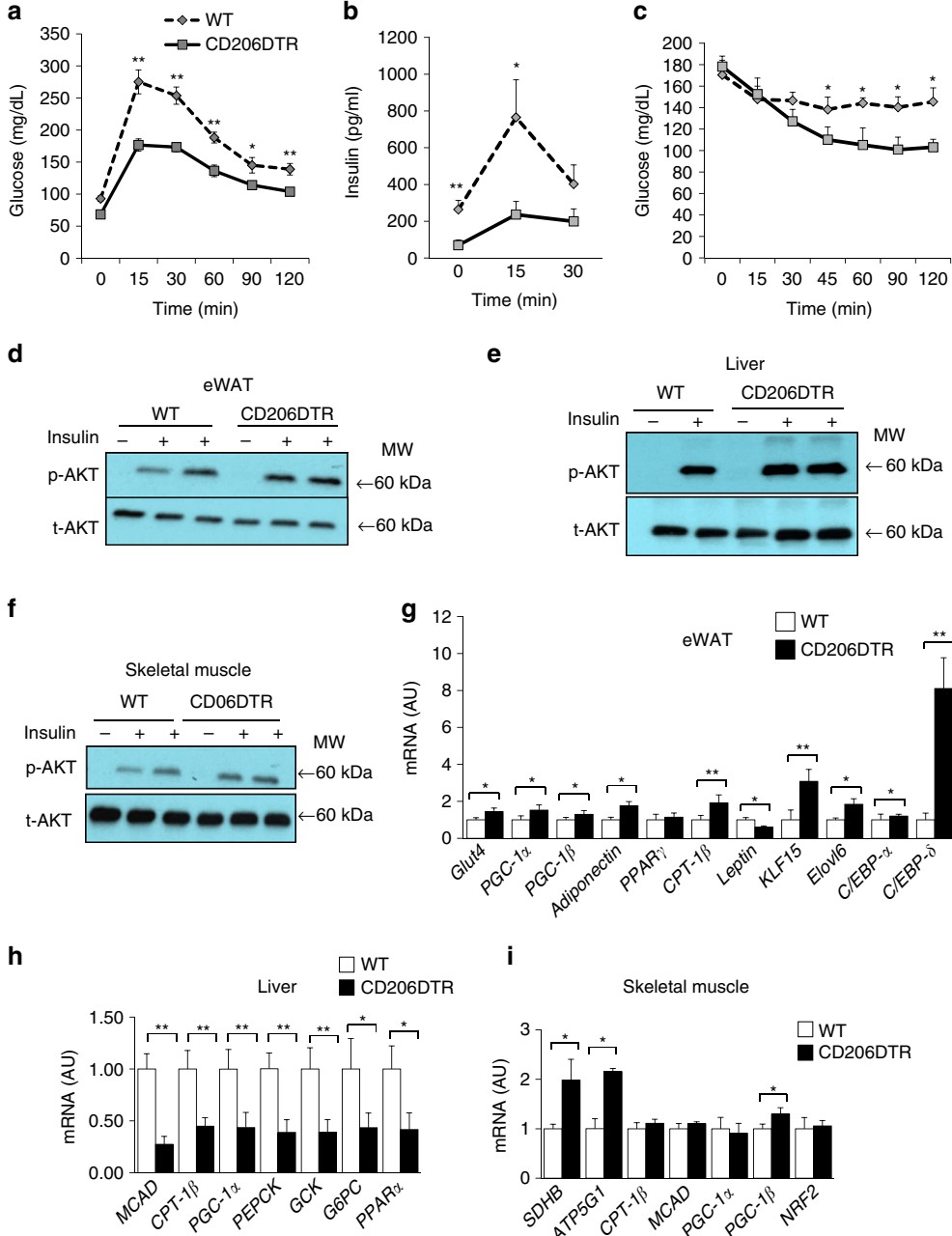

**Fig. 7** CD206[+] M2-like ATMs depletion promote insulin sensitivity in lean mice. **a, b** Glucose and insulin concentrations in an intra-peritoneal glucose tolerance test (IP-GTT), and **c** the glucose concentrations in an intra-peritoneal insulin tolerance test (IP-ITT) in NC-fed mice after DT treatment (*n* = 4–6 per group). **d–f** Western blot analysis of Akt phosphorylation in the eWAT, liver and skeletal muscle of DT-treated CD206DTR and WT littermates. (MW = Molecular weight). **g** Relative mRNA expression of mitochondrial, smaller adipocytes and adipogenesis marker genes in the eWAT (*n* = 3–5 per group). **h** Relative mRNA expression of gluconeogenesis and fatty acid oxidation genes in the liver and **i** skeletal muscle (*n* = 3–5). The data are shown as the means ± SEM. *$P < 0.05$, **$P < 0.01$ compared with littermates by Student's *t*-test

**Depletion of CD206[+] M2-like ATMs promote browning of white adipocytes.** Finally, we addressed whether depletion of CD206[+] M2-like macrophages affect beige progenitors. DT-treated CD206DTR and WT mice after cold exposure for 7 days resulted in markedly increased expressions of *UCP1*, as well as other browning marker genes in the iWAT[22, 37–39] (Fig. 10a). Immunofluoresence analysis also revealed increased UCP1 in CD206DTR mice upon cold exposure (Fig. 10b). Consistent with this, the beige progenitor marker genes were also upregulated (Fig. 10a). Interestingly, flow cytometry analysis revealed an increase in the PDGFRα/Sca-1 double positive cells after cold exposure in the iWAT of the CD206DTR mice compared with

WT control (Fig. 10c and Supplementary Fig. 12b). These data suggested that CD206[+] M2-like macrophages might be involved in regulating the browning of iWAT under cold stimulation. Further studies are required to investigate the role M2-like ATMs ablation in regulating the biological properties of beige progenitors and how they interact with each other to form niche will significantly enrich our knowledge.

**Discussion**
Several lines of evidence have suggested that ATMs are involved in maintaining insulin sensitivity in adipocytes through their

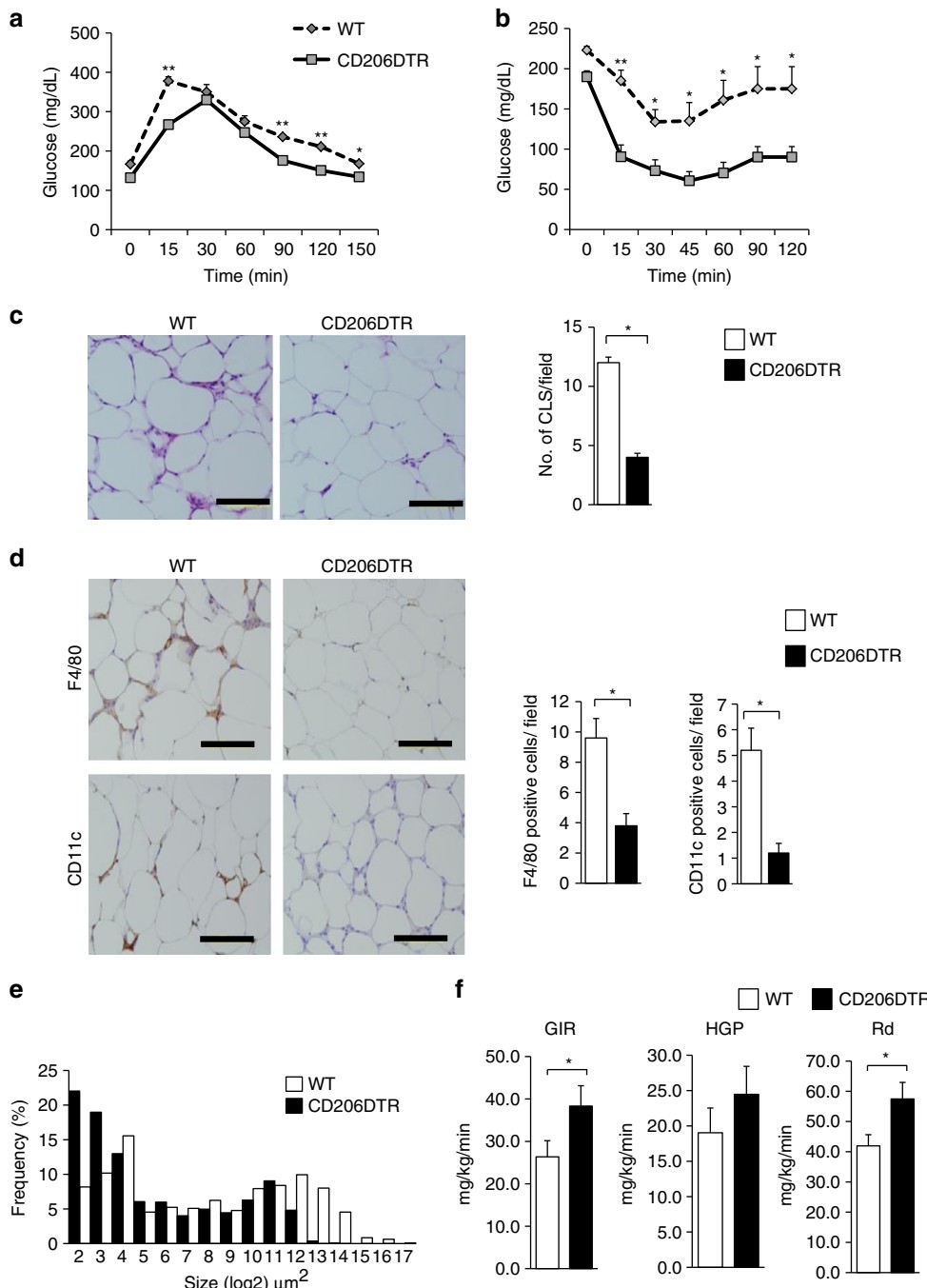

**Fig. 8** CD206[+] M2-like ATMs depletion promote insulin sensitivity in obese mice. **a** IP-GTT and **b** IP-ITT in 16-week HFD-fed mice after DT treatment (*n* = 5 mice per group). **c** HE staining of eWAT (*left panel*). The number of CLS/field is given in right panel. (*Scale bar*, 200 μm). **d** Representative images of a paraffin section of eWAT of the DT-treated HFD-fed CD206DTR and WT mice stained with anti-CD11c and anti-F4/80 antibody (*left panel*). The number of F4/80 and CD11c positive cells/field (*right panel*) (*n* = 3–4 mice per group) (*Scale bar*, 200 μm). **e** A histogram of adipocyte cell size in the eWAT of DT-treated HFD-fed CD206DTR mice compared with WT mice (*n* = 4–5 mice per group). **f** Hyperinsulinemic-euglycemic clamp study showed glucose infusion rates (GIR), hepatic glucose production (HGP), and rate of glucose disappearance (Rd) in HFD-fed DT-treated CD206DTR and WT mice (*n* = 9–10 mice per group). The data are shown as the means ± SEM. *$P < 0.05$, **$P < 0.01$ compared with littermates by Student's *t*-test

anti-inflammatory actions, in collaboration with other leukocyte lineages including Treg cells, eosinophils, and invariant natural killer cells[1, 2, 4, 22, 35]. In the current study, we determined that CD206 is an ideal marker to target M2-like macrophages in adipose tissues. Thus, we are able to specifically deplete M2-polarized macrophages in adipose tissue of DT-treated CD206DTR mice. So far, congenital deficiency of M2-like ATMs reportedly causes lipodystrophy-like pathophysiology

accompanied by accelerated lipolysis[40]. As insulin resistance and obesity related disorders progresses with advanced age, and M2-like macrophages are up-regulated in the early phases of obesity[34], it is important to deplete M2-like macrophages at a specific time point to examine the developmentally time sensitive role of M2-like macrophages. Thus, conditional and partial depletion have advantages over congenital or genetic ablation of M2-like macrophages. There are a number of reports

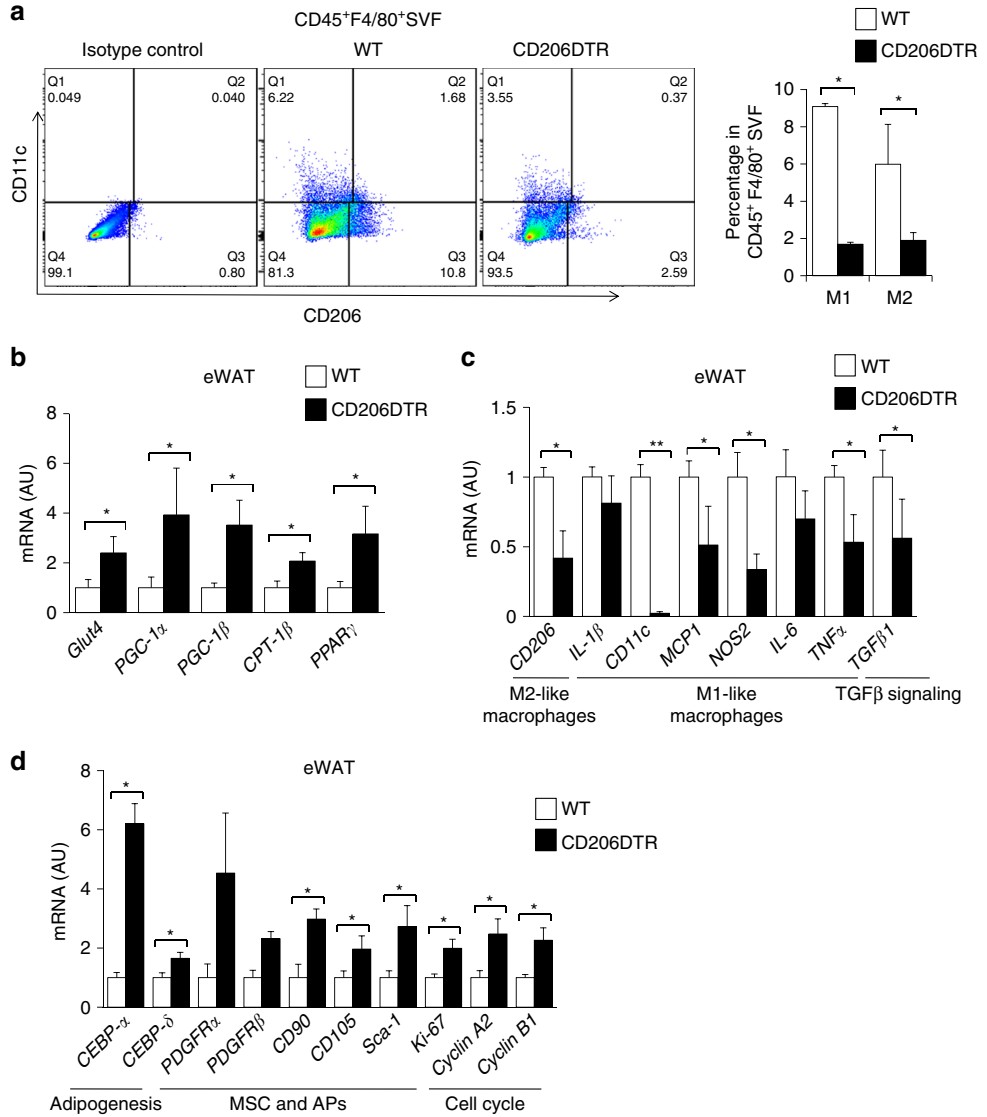

**Fig. 9** Effects of CD206+ M2-like ATMs depletion on adipogenesis in obese mice. **a** Representative flow cytometry analysis of CD11c and CD206 expression in F4/80+ SVF cells of eWAT from WT and CD206DTR mice. Cells were also stained with isotype control antibodies for anti-CD11c and anti-CD206 antibodies (*left panel*). Percentages of M1- and M2-like macrophages in CD45+F4/80+SVF of eWAT from WT and CD206DTR mice were calculated (*right panel*). Full gating strategy is given in Supplementary Fig. 13a. **b** The relative mRNA expression of metabolic genes and **c**, **d** M1-, M2-like macrophages, adipogenesis, APs/MSC and cell cycle markers genes in eWAT of DT-treated HFD-fed CD206DTR and WT mice. ($n = 3$–5 per group). The data are shown as the means ± SEM. *$P < 0.05$, **$P < 0.01$. (H) The data are shown as the means ± SEM. *$P < 0.05$, **$P < 0.01$ compared with littermates by Student's $t$-test

demonstrating that partial and conditional depletion of specific cell lineages can bring about particular biological events without affecting the functions of other cells lineages[41–45]. In this study, we have successfully shown that partial but specific depletion of CD206+ M2-like macrophages in adult mice improved glucose metabolism without affecting body weight, food intake, and the numbers and functions of other cell lineages, including M1 macrophages. It has been reported that cold stimulation promotes the differentiation of beige precursors into beige/brite adipocytes within one week of exposure[38, 46]. Recently, Chawla's group reported that eosinophil or M2 macrophages-derived type 2 cytokines, IL-4/IL-13, mediates cold stimulation-induced biogenesis of beige fat[2]. There are several reports indicating that M2 macrophages polarization regulates thermogenesis and browning of iWAT by increasing the expression of *UCP1, PGC-1α*, and other browning genes [2, 47, 48]. Previous studies indicated

that PDGFRα+ progenitors were recruited in WAT upon ADRB3 stimulation[26] and these APs were also involved in different compensatory mechanism involving cellular restoration and repair in WAT and other tissues[3, 49]. It is well documented that type 2 cytokines and ATMs were involved in regulation of APs remodeling[3, 22, 39, 48]. However, it is unknown how decline of CD206+ M2-like macrophages affect WAT upon cold stimulation. Here, we report that decline of CD206+ M2-like macrophages promoted browning of iWAT. Our findings suggest that CD206/TGFβ signaling play a critical role in maintaining APs in a quiescent state in an analogous manner to that of the non-myelinating Schwann cell-based niche, where hematopoietic stem cells (HSCs) are maintained in hibernation via TGFβ-dependent pathways[11]. As we have previously reported, M2-like macrophages expand, albeit mildly compared with M1-like macrophages, during the course of obesity progression[34].

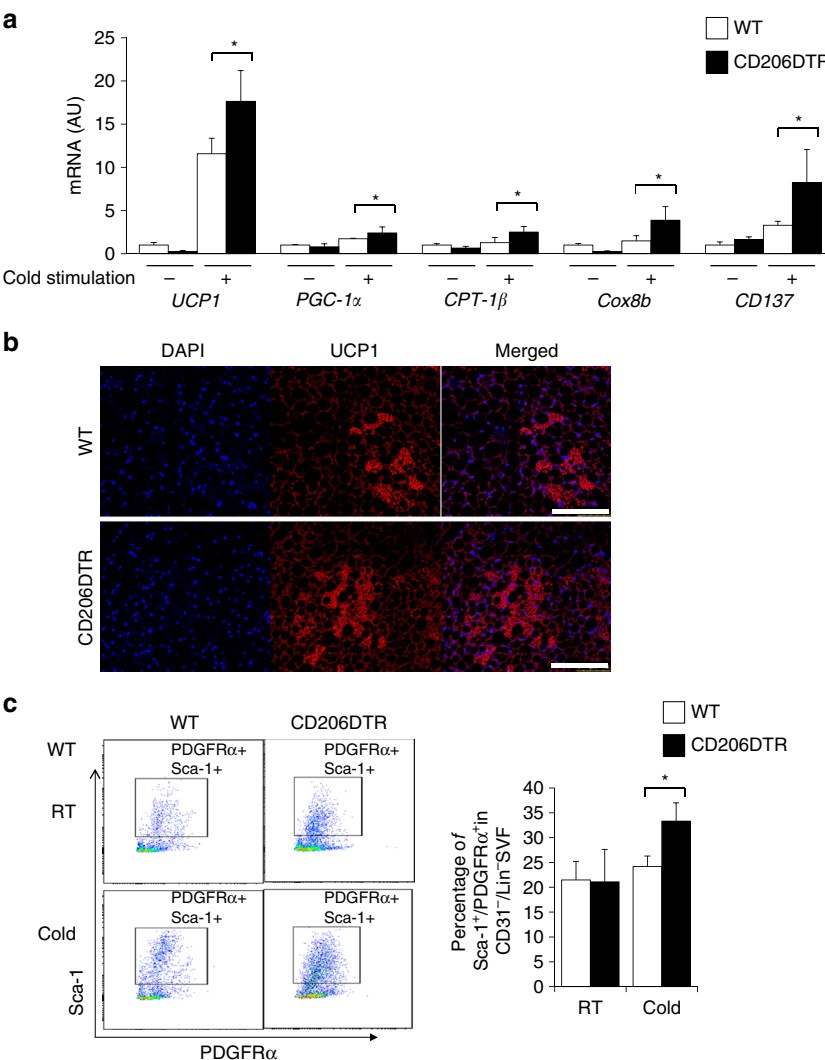

**Fig. 10** Depletion of CD206[+] M2-like macrophages promote browning of iWAT. **a** Relative mRNA expression of *UCP1, PGC-1α. CPT-1β* and *CD137* levels in iWAT after CD206[+] cells depletion under cold exposure (*n* = 3–5 mice per group). **b** Immunostaining of paraffin section of eWAT stained with anti-UCP1 antibody in DT-treated CD206DTR mice after cold exposure compared with WT control. The images were taken with a Leica TCS-SP5 (40×). **c** Representative flow cytometry analysis of PDGFRα/Sca-1 double positive cells from iWAT of DT-treated CD206DTR and WT mice at RT and cold stimulated (25 °C and 6 °C) (left panel). The percentage of Lin(−)APs positive fraction in the DT-treated CD206DTR mice are shown in the right panel (*n* = 3–5). For this, first, negative selection of PE-Cy7 anti CD31 (endothelial), FITC anti-mouse lineage cocktail were selected followed by positive selection of PE anti-PDGFRα and APC-Cy7 anti Sca-1. Full gating strategy is given in Supplementary Fig. 13b.The data are shown as the means ± SEM. *P < 0.05 compared with littermates by Student's *t*-test

Therefore, it may be possible that the mild expansion of M2-like macrophages during the progression of obesity hampers APs proliferation and subsequent adipogenesis. Thus, the limited adipogenesis due to expanded M2-like ATMs may increase the lipid burden of the existing fat cells, inducing hypertrophy rather than hyperplasia of adipose tissues, thus impairing insulin resistance. A series of these events may create a vicious cycle, accelerating the progression of obesity-associated metabolic disorders. In this regard, it is expected that CD206[+] M2-like macrophage-targeting drugs may exert effective therapeutic effects by breaking the vicious cycle between M2-like macrophages and APs. As reduction of CD206[+] M2-like macrophages did not affect systemic adiposity, CD206[+] M2-like ATMs-targeting drugs would not promote obesity, as is observed in the case of PPARγ activators (e.g. thiazolidinedione).

In HFD-fed mice, both CD11c[+]CD206[-] and CD11c[+]CD206[+] ATMs are markedly increased; DT-treatment depleted not only CD206[+]CD11c[-] but also CD206[+]CD11c[+] ATMs. Although it seems that CD206[+] M2-like ATMs plays a major role in the insulin-sensitive phenotype associated with reduced inflammation, the possibility that depletion of CD206[+]CD11c[+] ATMs contributes to this phenotype cannot be completely excluded.

In conclusion, the current study demonstrates a new role for CD206[+] M2-like macrophages in constituting a microenvironment for APs, in a TGFβ-dependent manner, to retain systemic insulin sensitivity by tuning the quiescence/proliferation balance of APs to adapt to changes in nutritional status. Our findings suggest a new strategy for the development of drug discovery for the treatment of insulin resistance and type 2 diabetes.

## Methods

**Materials**. Diphtheria toxin (Sigma, cat# D0564,) and collagenase (Sigma, cat# C6885) were purchased from Sigma-Aldrich (St. Louis, MO). Anti-human

diphtheria toxin receptor (anti-HB-EGF) was obtained from Cosmo Bio, cat# 71–503 (Tokyo, Japan), purified mouse anti β-catenin (Cat# 610154) from BD Bisocience, anti-rabbit Akt (Cat# 9272 S) and anti-rabbit p-Akt (S473) (Cat# 9271 S) were purchased from Cell Signaling for western blot analysis. The horseradish peroxidase linked anti-mouse, cat# NA931V and anit-rabbit, cat# NA934V secondary antibody was obtained from Amersham Bioscience (Buckinghamshire, UK). The ECL^TM western blot detection reagent was obtained from GE Healthcare, cat# RPN2232 (Buckinghamshire, UK). Tamoxifen was purchased from Sigma, cat# T5648 and bromodeoxyuridine (BrdU) was purchased from BD Biosciences, cat# 550891. D-Glucose-6,6-d2 was purchased from the Santa Cruz Biotechnology, cat# sc-257287A.

The PCR primers that were used with the TaqMan method were purchased from Applied Biosystems (Foster City, CA), and those used with the SYBR Green method were purchased from Invitrogen^TM Life Technologies, Japan (Tokyo, Japan). The SYBR Green primer sequence is given in Supplementary Table 1.

For the flow cytometry analysis, hamster CD11c conjugated with PE (Cat# 553802), PE rat anti-mouse SiglecF (Cat# 562068), purified rat anti-mouse CD16/CD32 (Cat# 553141), 7-amino-actinomycin D [7AAD] (Cat# 559925), PE ArHam IgG1 λ2 isotype control (Cat# 553954), FITC BrdU flow kit (Cat# 559619), APC rat anti-mouse TER-119/erythroid cell antibody (Cat# 557909) anitbodies were obtained from BD Biosciences. PE hamster rat polyclonal isotype control (Cat# ab32662-100) was purchased from Abcam. The rat anti-mouse CD206 conjugated alexa fluor 647 antibody (Cat# MCA2235A647) and the rat IgG2a alexa fluor 647 isotype antibody (Cat# 1212A647) were obtained from AbD Serotec (Oxford, UK). The APC anti-mouse CD206 (MMR) antibody (Cat# 141707) and the APC rat IgG2a, κ isotype control antibody (Cat# 400511), FITC anti-mouse Ly6G/Ly-6C (Gr-1) (Cat# 108406), FITC conjugate rat IgG2a isotype control, FITC anti-mouse CD3/Gr-1/CD11b/CD45R(B220)/Ter-119 (FITC lineage cocktail) (Cat# 78022), FITC Arm Hamster IgG/rat IgG2b/rab IgG2a isotype control (Cat# 78023), APC anti-mouse CD140a (Cat# 135908), PE anti-mouse CD140a (Cat# 135905), PE rat IgG1, κ isotype control (Cat# 553925) and APC/Cy7 anti-mouse Ly-6A/E (Sca-1) (Cat# 108125), the APC/Cy7 rat IgG2a, κ isotype control (Cat# 400523), rat anti-mouse F4/80 conjugated APC/Cy7 (Cat# 123117) antibodies were purchased from BioLegend. Anti-mouse CD45 PE-Cy7 (Cat# 25–0451), PE-Cy7 rat IgG2a, κ isotype control (Cat# 25–4321-81) and anti-mouse CD31 PE-Cy7 (Cat# 25–0311), APC-eFluor 780 anti-mouse CD11b (47-0112-82) antibodies were purchased from eBioscience.

For the immunohistochemistry experiments, the anti-human diphtheria toxin receptor (anti-HB-EGF) antibody was obtained from Antibodies—online (Cat# ABIN701052); the biotin anti-BrdU (Cat# 339809) and FITC anti-mouse F4/80 (Cat# 123107), APC/Cy7 anti-mouse F4/80 (Cat# 123107), the anti-TGFβ1 (Cat# sc-146), monoclonal anti-CD206 (Cat# sc-58987), and anti-rabbit perilipin (Cat# sc-67164) antibodies were obtained from Santa Cruz Biotechnology (Dallas, Texas); the anti-rabbit Phospho Smad2/3 (Cat# 8828) primary anitbody, anti-rabbit p27 (Cat# 3698) primary anibody, goat anti-rabbit IgG (H + L), alexa fluor® 555 conjugate (Cat# 4413), anti-rabbit alexa fluor 488 (Cat# 4413), and goat anti-rat IgG (H + L) alexa fluor® 488 conjugate (Cat# 4413) secondary antibodies were obtained from Cell Signaling Technology (Denvers, MA); and goat anti-guinea pig IgG (H + L) alexa aluor® 488 conjugate (Cat# A11073), donkey anti-rabbit IgG (H + L) alexa fluor® 488 conjugate (Cat# A21206) secondary antibodies and Click-iT® EdU alexa fluor® 647 Imaging Kit (Cat# 10340) were obtained from Life technologies. Anti-rabbit Ki-67 antibody was obtained from Abcam (Cat# ab15580) (Cambridge, UK). The anti-goat PDGFRα antibody (Cat# GT15150) was purchased from Neuromics. Anti-GFP (guinea pig) (Cat# GFP-GP-AF1180) were purchased from Frontier Institute Co.,Ltd. Cell mask green plasma membrane stains were purchased from molecular probes by Life Technologies, cat# C37608.

For the in vitro experiments, the Dulbecco's modified Eagle's medium was purchased from Gibco^TM Life Technologies, Japan (Tokyo, Japan); the MesenCult MSC basal medium (Mouse), MSC proliferation supplement (mouse), and the adipogenic stimulatory supplement (mouse) were purchased from STEMCELL (Vancouver, Canada); the recombinant Interleukin-4 (IL-4) (Cat# 214-14) was purchased from Peprotech (Rocky Hill, NJ); the prostaglandin E2 (PGE2) (Cat# 363-244-6) was purchased from Caymen Chemical (Ann Arbor, Michigan); the recombinant mouse TGFβ1 (Cat# 7666-MB-005), macrophage colony-stimulating factor (M-CSF) (Cat# 416-ML-050), and the monoclonal mouse anti-TGFβ1, 2, and 3 (Cat# MAB1835) were purchased from R&D Systems (Minneapolis, MN); the TGFβRI/II inhibitor (LY2109761) (Cat# CS-0571) was purchased from Chem Scene (Monmouth Junction, NJ).

**Generation and maintenance of mice**. To generate CD206DTR mice, we obtained mouse BAC clone RP 24-377 D19 carrying a 152-kbp insert containing the exon coding translational start Met and the upstream 133-kbp sequence of the CD206 gene from the BACPAC Resources Center CHORI (Oakland, CA). The plasmid pTRECK6, which includes a noncoding exon and intron from rabbit β-globulin gene, human HB-EGF (DTR) complementary DNA (cDNA), and rabbit β-globulin and simian virus 40 polyadenylation signals, was kindly provided by Dr. Kenji Kohno (Nara Institute of Science and Technology)[50, 51]. Using a Counter-selection BAC modification kit (Cat# K002) (Gene Bridges, Dresden, Germany), we removed 6-bp nucleotides (5′-GTTATG-3′; ATG corresponding to translational start Met of

CD206) and inserted a 2.3-kbp fragment containing part of the noncoding exon and intron of the rabbit β-globulin gene, DTR cDNA, and the polyadenylation signals from pTRECK6 to yield the BAC vector pTg-CD206DTR. The purified BAC DNA was microinjected into pronuclei of fertilized one-cell embryos from C57BL/6 mice by UNITECH (Chiba, Japan). The transgenic founders were then again backcrossed to C57BL/6 mice. The male F4 generations and beyond were used for experiments to derive the data. Wild-type littermates were used as controls in all the experiments.

To generate CD206-CreER^T2 mice, we created the targeting construct. The plasmid pCAG-CreER^T2 was obtained from Addgene (# 14797). We replaced the human DTR DNA fragment in the pTg-CD206-DT receptor BAC transgene[17] with mutated estrogen receptor fused Cre recombinase (CreER^T2) DNA fragment in the pCAG-CreER^T2 by using a Counter-selection BAC modification kit (Gene Bridges, Dresden, Germany) to yield the pTg-CD206-CreER^T2 (CD206-CreER^T2)[52]. In one series of experiments, we crossed male CD206-CreER^T2 mice with female TGFβ^flox/flox to obtain CD206-CreER^T2/ TGFβ^flox/flox. Male CD206-CreER^T2/ TGFβ^flox/flox and control littermate TGFβ^flox/flox were administered tamoxifen (0.15 mg/gBW) for four consecutive days. Male Pdgfrα-CreERT2-Egfp (PRa) mice were obtained from department of pathology, University of Toyama[53]. Mice were maintained under standard 12 h light and dark cycles. Male mice aged 8–12 weeks were used for all the experiments; the mice were allowed ad libitum access to water and standard chow (Nosan Corporation, Yokohama, Japan). For body weight and food intake measurements, the mice were caged individually. All the animal studies were conducted at the animal facility center of the University of Toyama. Animal care and procedures were approved by the Animal Experiment Committee of the University of Toyama (Authorization No. S2009 Med-41).

**Genotyping**. Whole genomic DNA was derived from the tail after lysing with DirectPCR(Tail) lysing solution from Viagen (Los Angeles, CA), according to the manufacturer's instructions. This crude DNA was then used for PCR using the Tks Gflex DNA Polymerase kit from TaKaRa (Shiga, Japan), according to the manufacturer's instructions and the following PCR conditions: one cycle of 95 °C for 5 min, 35 cycles of 94 °C for 30 s, 60 °C for 30 s, and 72 °C for 30 s, and one cycle of 72 °C for 7 min. The primers used for genotyping were purchased from Invitrogen^TM Life Technology (Tokyo, Japan) and had the following sequences: primer 1, TGTATTCTTTGCCTTTCCCAGTCTC (CD206 primer); primer 2, CCTCAA AACAGACTTACCCAATAGCTG (CD206 primer); primer 3, AAGAGGAGACAATG GTTGTCAACAG (DTR specific primer). The PCR products were subsequently separated using 1.5% agarose gel electrophoresis for 30 min. The DNA was visualized in the gel by the addition of ethidium bromide (1:1000 dilution) to the gel solution. The expected sizes of the DNA fragments for CD206 and CD206 DTR were 138 bp and 257 bp, respectively.

**Realtime polymerase chain reaction (RT-PCR)**. Tissues for RT-PCR were collected and preserved in RNA later solution from Ambion (Austin, Texas) according to the manufacturer's instructions. Tissue RNA was extracted using an RNeasy kit, cat# 74106 (Qiagen, Hilden, Germany) and was reverse transcribed using TaKaRa PrimeScript RNA Kit, cat# RR036A (Shiga, Japan), according to the manufacturer's instructions. Quantitative PCR of the genes was performed using the TaqMan method (1 cycle of 50 °C for 2 min, 95 °C for 10 min, and 40 cycles of 95 °C for 15 s, 60 °C for 1 min) using premade primer sets. The relative mRNA expression levels were calculated using the standard curve method and were normalized to the mRNA levels of 18 S or *TF2B*. The SYBR Green thermal cycling conditions were 1 cycle of 95 °C for 30 s, and 45 cycles of 95 °C for 10 s and 60 °C for 20 s. The relative mRNA expression levels were calculated using the standard curve method and were normalized to the mRNA levels of *β-Actin* or *TF2B*.

**Diphtheria toxin injection**. Diphtheria toxin was diluted with sterile PBS(−) to the desired concentrations and was intraperitoneally injected at a dose of 3 ng/gram body weight (low doses) 3–4 times every other day. The experiments and procedures were performed 2 days after the last injection.

**Western blot**. Tissues for the western blot analysis were quickly frozen in liquid nitrogen and preserved at −80 °C until utilization. The western blot analysis was performed as described previously with a slight modification. Briefly, the tissues for western blotting were homogenized in lysis buffer containing 25 mM Tris–HCl (pH 7.4), 10 mM Na$_3$VO$_4$, 100 mM NaF, 50 mM Na$_4$P$_2$O$_7$, 10 mM EDTA, 0.2% leupeptin (5 mg/mL), 0.5% aprotinin (5 mg/mL), 2 mM phenylmethylsulfonyl fluoride, and 1% Nonidet P-40 using a Multi-Beads Shocker^TM cell disrupter. The lysates were centrifuged to remove any insoluble materials. The lysates were mixed with loading buffer before protein denaturation by boiling at 95°C for 3–5 min.The protein lysates were run on 7.5% separating gels and transferred afterwards to PVDF Immobilon-P transfer membranes (Millipore, Billerica MA). The membranes were incubated overnight at 4°C for the primary antibody (1:1000 dilution) and for 1 h at room temperature for the second antibody (1:2000 dilution) before being subjected to a western blot detection reagent immediately before image development.

**Immunohistochemistry**. Paraffin-embedded tissues were cut 5-μm thick and mounted on slides. Slide staining with the first and second antibodies was performed according to the manufacturer's instructions. A specific cell count was performed for at least three randomly chosen 200 × magnification fields. The average adipocyte size was measured by dividing the surface area of a 100 × magnification field by the number of adipocytes. For BrdU staining, the mice were injected with 2 mg of BrdU 2 h later, the mice were sacrificed and the tissues were collected for immunohistochemistry.

In vitro immunohistochemistry was performed by growing the cells on an 18-mm round cover glass in 6-well dishes. The cover glasses were thin-coated with rat tail Collagen I from Gibco™ Life Technologies Japan (Tokyo, Japan) before cell culture. A cell specific count was performed in a 100 × magnification field.

For frozen section, eWAT from mice collected in 4% PFA after systemic perfusion. The tissues were kept at room temperature for 2–3 h. After that tissues were incubated in PBS(−) for one overnight and 30% sucrose one overnight in shaker at 4 °C. At the end the tissues were placed in block by adding OCT compound and immediately kept block at −80 °C for at least 24 h to solidify it. Then the frozen tissues were cut into 15–20 μm thickness by using cryostat. EdU injection (20 nM/mice) was injected 2 and 96 h before sacrifice the mice. After making the frozen block, immunofluoresence staining was performed by using EdU Alexa 647 vs. anti-goat PDGFRa, 1:50 dilution (Neuromics) and anti-perilipin 1:50 dilution, ant-rat CD206, 1;50 dilution and anti-rabbit TGFβ, 1:50 dilution followed by relevant secondary antibodies (1:250 dilution). All micrographs were taken with Keyence BZ-8000, Olympus BX6, or TCS SP5 Leica confocal microscopes (Oil 63×), and image processing was performed with the BZ-Analyzer software.

**Flow cytometry analysis**. Isolation and separation of the SVF and subsequent flow cytometry were performed as previously described[34, 54, 55]. Cells in the SVF of eWAT, after exclusion of dead cells by gating with 7AAD, live cells were selected for further analysis. M1-or M2-like macrophages were identified as CD45-positive/ F4/80-positive/ CD11c-positive/ CD206-negative or CD45 positive/ F4/80-positive/ CD11c-negative/ CD206-positive cells, respectively (1:50 dilution). The flow cytometry detection of the APs was performed similarly to a previous method[3, 56]. First, negative selection of CD31+ (endothelial) (1:100), FITC-lineage cocktail cells (1:100) were selected followed by positive selection of PDGFRα+ (1:50) and Sca-1+ cells (1:50). These experiments were performed with a FACSDiva Version 6.1.2 automated cell analyzer (Becton Dickinson FACSCanto II) and cell sorting was performed by an automatic cell sorting analyzer (Becton Dickinson FACSAria SORP). Flow cytometry analysis for detection of BrdU intake was performed after the mice were injected with BrdU (10 mg/kg body weight) 3–4 times every other day at the same time as the DT injections. The SVF preparation and BrdU detection were performed using an FITC BrdU flow kit (BD Pharmingen™; BD Bioscience, San Diego, CA) according to the manufacturer's instructions. Live cells were gated for CD45+ and CD45- population. Then BrdU was analyzed in both fraction including CD45+BrdU+ and CD45−BrdU+ population. Data analyses were performed "offline" using FlowJo software. Unstained specimen and isotype negative control were used for all relevant samples to justify gating strategy. Fluorescence minus one (FMO) controls was used wherever needed.

**Magnetic activated cell sorting study**. eWAT was dissociated into SVF to isolate APs as previously described[22, 54]. The SVF were processed for magnetic sorting with anti-CD31, anti-CD45 and anti-Sca-1 microbeads. MicroBead Kit mouse were purchased from Miltenyi Biotech. First, negative selection with anti-CD31 microbeads (Cat# 130-097-418) and anti-CD45 microbeads (Cat# 130-052-301) was conducted. This negative fraction was then incubated with an anti-Sca-1 microbead kit (FITC) (Cat# 130-092-529). The purified cells were subjected to RNA extraction and qPCR analysis of cell proliferation markers. All incubations and dilution were performed at 4°C for 10- 20 min according to the manufacturers' instructions. The enriched (positive) fraction was then centrifuged and subjected to RNA extraction and qPCR analysis.

**In vitro co-culture**. The BMDM and SVF were isolated as previously described[10]. BMDMs were cultured in DMEM with the addition of M-CSF (100 ng/mL) for one week and were induced with IL-4 (10 ng/mL) and PGE2 (50 ng/mL) 24 h before co-culture with ASCs. The ASCs were derived from an iWAT SVF cultured with complete MesenCult proliferation medium that was prepared and used according to the manufacturer's instructions (Stemcell Technologies, Cat# 05501). Briefly, for the adipose stem cell proliferation, the SVF, after the red blood cell lysis, were cultured with complete Mesencult proliferation medium, a combination of Mesencult MSC basal medium and MSC stimulatory supplement (Stemcell Technologies, Cat# 05502) in a 1:4 volume ratio, in a 10-mL dish with 1–2 million cells per dish. 1 day after seeding, the old medium was changed to new medium to remove the non-attached cells. The attached cells were grown to 60–70% confluence before co-culturing with ASCs or for further re-seeding. To assess the inhibitory effect of the BMDMs on ASC proliferation, a low number of BMDMs and ASCs (15,000–30,000 cells) were seeded together in 6-well dishes to avoid cell growth arrest because of early confluence. After 1–2 days of co-culture, the cells

were collected for PCR or used for immunohistochemistry. A TGFβRI/II inhibitor and anti-TGFβ antibodies were added to the medium at final concentrations of 5 ng/mL, and 0.5 μg/mL, respectively, on the same day the ASCs and BMDMs were seeded.

To assess the inhibitory effect of the BMDMs on adipogenesis, the ASCs and BMDMs were cultured in equal numbers (100,000 cells) in 6-well dishes. ASCs alone and ASCs with the addition of recombinant mouse TGFβ1 (5 ng/mL, final concentration) were used as controls. Recombinant mouse TGFβ1 and a TGFβRI/II inhibitor were added on the first day of co-culture in Mesencult complete MSC stimulatory medium. After 2 days of confluence, the medium was changed to Mesencult complete adipogenic stimulatory medium, which was a combination of Mesencult MSC basal medium and Mesencult adipogenic stimulatory supplement with a 1:4 volume ratio. Every 4–5 days, the medium was changed with new complete stimulatory medium with the addition of new reagents (Recombinant Mouse TGFβ1, TGFβRI/II inhibitor, and M-CSF). After 12 days of adipogenesis induction, the cells were collected for PCR and Oil Red O staining (Thermo Scientific, SC 00011) for adipocyte visualization.

**Glucose tolerance test and insulin tolerance test**. In the intraperitoneal glucose tolerance test (IP-GTT), 6 h fasted DT-treated CD206DTR and littermate were injected with glucose (1 mg/g BW) intraperitoneally. In the intraperitoneal insulin tolerance test (IP-ITT), 2 h fasted mice were injected intraperitoneally with human insulin 0.8 units/kg BW for NC-fed and 1.2 units/kg BW for HFD-fed mice. Blood samples were then collected 0,15, 30, 45, 60, 90 and 120 min from the tail vein. The blood glucose levels were measured using the STAT STRIP Express 900 (Nova Biomedical, Waltham MA). The serum insulin levels were determined using the Mouse Insulin ELISA KIT (ARKIN-031, Shibayagi, Shibukawa, Japan).

**Hyperinsulinemic-euglycemic clamp study**. Clamp studies were performed on 16 weeks HFD-fed DT-treated CD206DTR and WT littermate under conscious and unstressed condition after 6 h fasting as described previously[57, 58]. A primed-continuous infusion of insulin (Humulin R; Lilly) was given 10.0 milliunits/kg/min for HFD-fed mice, and the blood glucose concentration, monitored every 5 min, was maintained at ~120 mg/dl by administration of glucose (50% glucose enriched to ~20% with 50% D2-glucose (Sigma) (4:1) ratio for 120 min. Blood sample collected at 0, 90, 105, and 120 min for determination of the rate of glucose disappearance (Rd), and hepatic glucose production (HGP) was calculated as the difference between Rd and exogenous glucose infusion rates (GIR)[57].

**Statistical analysis**. Statistical analyses were performed using unpaired Student's t-tests or ANOVAs with the post Tukey–Kramer test and Bonferroni correction. Differences were considered statistically significant at *$P < 0.05$, **$P < 0.01$. The results are presented as the means ± SEM.

**Data availability**. The data supporting the findinds of the study are included in the Figures and Supplementary Information or can be obtained from the authors upon reasonable request.

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

## Acknowledgements

We would like to thank Prof. Kenji Kohno (Nara Institute of Science and Technology, Nara, Japan) for providing the pTRECK6 vector. The plasmid pCAG-CreERT2 was a gift from Connie Cepko (Addgene plasmid # 14797). This work was supported by Grants-in-Aid for Scientific Research from the Ministry of Education, Science, Sports, and Culture, Japan (26461327 to K. Tobe, 22790853 to S.F. and 22590971 to I.U.), by the Novonordisc Insulin Foundation to S.F. and a Lilly research grant to S.F., and by the grant for National Center for Global Health and Medicine (26A105) to T.O.

## Author contributions

A.N. and A.A. performed the experiments, acquired, and analyzed the data, and wrote the manuscript. T.K., T.I., A.T., Y.I., M.I. and S.F. acquired and analyzed the data (flow cytometry). S.Y. and K.T. acquired the immunostaining data. S.Y. performed the experiment to analyze fibrosis of eWAT. Y.N. helped in western blot experiment. M.S., J.I., I.U., K.K., Y.N., and S.F. analyzed and interpreted the data. A.A., T.K., Y.I., S.S., and T.N. provided assistance to perform the experiments, animal care and revised the article. K.T., and A.Ando. analyzed and interpreted the clamp data. H.M., provided help to generate the transgenic mice. A.A., S.F., and H.M. generated CD206DTR mice. T.K.,

H.M., and T.O. generated CD206-CreERT2 mice. K.U., Y.O., K.T., T.N., M.S., M.M., K.N., and K.S. revised the manuscript. K.T. supervised the project. All authors approved the final version of the paper for publication.

## Additional information

**Competing interests:** The authors declare that they have no competing financial interests.

