## [Peer Review File · Nature Communications]

Reviewers' Comments:

Reviewer #1 (Remarks to the Author)

Nawaz et al present data supporting the model where CD206+ Adipose tissue macrophages (ATMs) suppress adipogenesis via TGFB secretion. They primarily rely on ablation models using CD206 specific transgenic mice previously developed.

The story is compelling and interesting - however the data presented has many gaps that prevent me from fully assessing the validity of the story and falls short of me being convince that all the metabolic changes are due only to ATMs given that macropahges in many other metabolic organs are also effected.

Major issues:

1. Fig 1a-b - it is highly likely that the F4/80+ CD206- population are dendritic cells to explain the low F4/80 gene expression. The authors should examine DC genes in this analysis. The concept that these cells have "low viability" is speculative at the moment and can be tested with viability stains.
2. Fig 1g - the flow cytometry is unclear the degree to which CD206+ ATMs are lost in the DTR mice. It looks quite modest and doesn't seem to match what is seen in Fig 1a. This should be regated similar to 1a to demonstrate the true degree of knockdown. The y axis "%" should be more clearly labeled - % of what? The legend states that the control are "green dots" but these aren't apparent in the figure.
3. The authors do not state the duration of the DT treatment in any of the experiments - therefore it is very hard to understand any of their conclusions. The control is not well described as well - were they WT mice injected with DT?
4. The overall model is at odds with resident macrophage deficient models such as the Trib1ko mouse and the Csf1 knockout mice which do not demonstrate any increase in adipogenesis or AP. How the authors explain this or put thier results in this context.
5. Fig 4d appears to show a GFP+ CD206+ cell with the use of PDGFRA-CreER mice - this is concerning for off targeting of PDGFRA and does not support the conclusion from this data.
6. Fig 7 shows improvement in GTT and ITT in chow fed CD206DT mice. It is unclear if this is due to adipose tissue or not. It is more likely that this may relate to liver funciton and the deletion of kupffer cells. It is unclear how the adipose tissue phenotype may generate this metabolic phenotype in animals that are already insulin sensitive at baseline.
7. For Fig 8C - this is uninterpretable without some assessment of the efficiency of TGFB ablation with the TAM treatment in these mice.
8. Fig 9 which is treatment of HFD fed mice with CD206 ablation are uninterpretable without showing the degree to which CD206+ are ablated in the model. The data shown in Fig 10a are unclear how they defined the gates that they did. There appear to be CD11c+CD206+ double positive cells - are those the ones that are actually targeted. The use of fluorescence minus one controls would make me feel more comfortable that their gating is justified.
9. A critical gene to examine in Fig 9-10 is TGFB1 - is that suppressed to relate to the mechanism of increase in adipogenic regulated genes?

Reviewer #2 (Remarks to the Author)

The present study provides compelling evidence showing that CD206+ macrophages in adipose tissues play a role in adipocyte progenitor proliferation and whole-body glucose homeostasis. The authors firstly found that CD206 expression was enriched in M2 macrophages in adipose tissues. Depletion of CD206+ cells by DTR stimulated cell proliferation in adipocyte progenitors and also improved glucose homeostasis in vivo. The authors also showed that these actions were mediated

through a down-regulation of TGF- β signaling. Overall the data are robust that it clearly suggests the biological importance of CD206⁺ macrophages in adiposity and glucose homeostasis. I would suggest the following points that the authors wish to address to strengthen the authors' conclusions further.

1. TGF- β in adipose fibrosis has been shown to promote adipose fibrosis. Given the role of CD206⁺ macrophages in TGF- β signaling, the authors should investigate if adipose fibrosis is altered by DTR-depletion of CD206⁺ cells as well as by TGF- β deletion in CD206⁺ cells.
2. The role of M2 macrophages in beige adipocyte development has been shown previously. The authors should carefully analyze the browning phenotype in DTR-depletion of CD206⁺ cells. In addition, the data that no difference in body weight seems inconsistent with the role of M2 macrophages in promoting beige fat development; the authors wish to comment on the apparent discrepancy.
3. Considering the robust changes in systemic glucose tolerance in CD206⁺ cells, it would be insightful to determine the tissues in which glucose uptake is enhanced by CD206-DTR.
4. Concerning the data in Fig. 1h, the authors wish to comment on several M2 markers are disproportionately decreased by CD206-DTR.

Point-by-point Responses to the comments by Reviewers

Reviewer #1 (Remarks to the Author):

Nawaz et al present data supporting the model where CD206+ Adipose tissue macrophages (ATMs) suppress adipogenesis via TGF β secretion. They primarily rely on ablation models using CD206 specific transgenic mice previously developed. The story is compelling and interesting - however the data presented has many gaps that prevent me from fully assessing the validity of the story and falls short of me being convince that all the metabolic changes are due only to ATMs given that macrophages in many other metabolic organs are also effected.

Major issues:

Reviewer's comment 1:

Fig 1a-b - it is highly likely that the F4/80+ CD206- population are dendritic cells to explain the low F4/80 gene expression. The authors should examine DC genes in this analysis. The concept that these cells have “low viability” is speculative at the moment and can be tested with viability stains.

Our response:

We repeated the flow cytometry analysis according to the Reviewer`s suggestion. We determined that the F4/80⁺CD206⁻ population expressed higher level of *CD11c*, *IL-6* and *TNF α* mRNA than the F4/80⁺CD206⁺ population, indicating that the former includes substantial amounts of M1 macrophages. Furthermore, gene expression and flow cytometric analyses showed that the F4/80⁺ CD206⁻ population did not express CD103, showing that these fraction was not DCs.

In our revised manuscript, we have replaced the words “low viability” by the following sentence “the F4/80⁺CD206⁻ population expresses CD11c, IL-6 and TNF α , indicating that this population includes M1 macrophages”.

Figure 1.

Supplementary Fig. 1b

Flow cytometric analysis of F4/80⁺ CD206⁻ population in the eWAT

Reviewer's comment 2:

Fig 1g - the flow cytometry is unclear the degree to which CD206⁺ ATMs are lost in the DTR mice. It looks quite modest and doesn't seem to match what is seen in Fig 1a. This should be regated similar to 1a to demonstrate the true degree of knockdown. The y axis "%" should be more clearly labeled - % of what? The legend states that the control are "green dots" but these aren't apparent in the figure.

Our response:

In our revised manuscript, flow cytometry analysis showed that significant depletion of CD206⁺ cells (approximately 50-60%) was achieved in DT-treated CD206DTR mice. We have regated the FACS data according to the suggestion by the Reviewer (Figure 1g). We have corrected Y axis as "% of CD45⁺ population". We also deleted the sentence that mistakenly appeared regarding green dots in figure legend in our previous manuscript. The gating strategy and the information regarding isotype controls were described in Supplementary Information.

Reviewer's comment 3:

The authors do not state the duration of the DT treatment in any of the experiments - therefore it is very hard to understand any of their conclusions. The control is not well described as well - were they WT mice injected with DT?

Our response:

In our revised manuscript, we have stated the duration of DT treatment in Fig 1c (right panel), which was also highlighted in the main text. The same protocol was used for the rest of the experiments regarding DT treatments.

Reviewer's comment 4:

The overall model is at odds with resident macrophage deficient models such as the Trib1ko mouse and the Csf1 knockout mice which do not demonstrate any increase in adipogenesis or AP. How the authors explain this or put their results in this context.

Our response:

The discrepancy in the phenotypes between CD206-reduced mice and Trib1 knockout or Csf1 knockout mice can be explained by the differences in the way and duration of depletion (i.e. conditional and transient *versus* genetic and permanent). In our CD206-reduced mice, M2 ATMs were *transiently* ablated in adult animals. As we showed in the current study, CD206⁺ cells provide a niche for APs to preserve the APs pool by keeping them in a quiescent state, which was supported by our observation that transient CD206⁺ cell ablation stimulated AP proliferation, and thus, promoted adipogenesis. By contrast, M2 macrophages are permanently ablated in Trib1 knockout and Csf1 knockout mice from the birth, indicating that AP niche function of CD206⁺ resident macrophages is constantly impaired. In NC-fed, the absence of AP niche may not create any abnormal structures in WATs due to adaptive responses in those mice. However, the absence of AP niche CD206⁺ cells promotes exhaustion of APs when the requirement of AP proliferation and adipogenesis has chronically been enlarged, for example, by HFD when adaptive adipogenesis has to occur to absorb excess nutrients. Consistent with this idea, lipodystrophic phenotypes of Trib1 knockout mice become overt only when they are fed by HFD^{1,2}. In any event, the strategy for *conditional* depletion has an advantage over genetically *permanent* depletion in that the former can provide the way to evaluate the “direct” and “physiological” effects of the ablation of specific lineage cells, excluding the effects of secondary phenomena and various adaptive responses.

Reviewer's comment 5:

Fig 4d appears to show a GFP⁺ CD206⁺ cell with the use of PDGFRA-CreER mice - this is concerning for off targeting of PDGFRA and does not support the conclusion from this data.

Our response:

In Fig. 4d, PDGFR α ⁺ cells are marked by eGFP (green), and CD206⁺ cells are stained by anti-CD206 antibody followed by anti-rat Alexa 555-conjugated secondary antibody. Our aim is to show that CD206⁺ cells are located in close vicinity to PDGFR α ⁺ cells; however, the finding obtained from Fig, 4d is rather suggestive but not convincing. Therefore, we replaced this figure into Supplementary information.

Reviewer's comment 6:

Fig 7 shows improvement in GTT and ITT in chow fed CD206DT mice. It is unclear if this is due to adipose tissue or not. It is more likely that this may relate to liver function and the deletion of kupffer cells. It is unclear how the adipose tissue phenotype may generate this metabolic phenotype in animals that are already insulin sensitive at baseline.

Our response:

We are really grateful to the Reviewer for his suggestion. We understand the concerns raised by the reviewer#1, whether insulin sensitivity was related to adipose tissue or not. This Reviewer presumed that insulin sensitivity may be related to the liver, probably due to depletion of Kupffer cells. In fact, the expression of CD206 gene (*mrc1*) is down-regulated in the liver of DT-treated CD206DTR mice (Supplementary Fig. 4f).

We performed hyperinsulinemic-euglycemic clamp studies to know which of the tissues, the skeletal muscle/ adipose tissue or the liver, contribute to improved insulin sensitivity. As shown in the Figure 9f in our revised manuscript, CD206⁺ cell depletion enhanced glucose infusion rate (GIR) while it did not affect insulin-induced inhibition of hepatic glucose production (HGP). Thus, skeletal muscle and adipose tissues play major roles in improving insulin sensitivity in CD206⁺ cell-depleted mice. Although DT treatment reduced Kupffer cells in the liver as shown above, it may not contribute much to improvement in the insulin-induced inhibition of HGP.

Thus, how adipose tissue phenotype changes in CD206⁺ cell-depleted mice are related to insulin sensitivity with increased GIR without affecting HGP? Adipose tissue alterations in CD206⁺ cells depleted mice is characterized by increased adipogenesis and upregulated expression of

adiponectin gene, which is similar to those in thiazolidinedione (TZD)-treated mice (Kubota N et al., 2006). In that report, TZD improved insulin sensitivity by stimulating the adipogenesis and upregulated expression of adiponectin genes like our DT-treated CD206DTR mice. In addition, their hyperinsulinemic-euglycemic clamp studies revealed increased GIR without affecting HGP compared to the non-treated animals, which is consistent with our clamp study result in CD206⁺ cell-depleted mice.

Even in lean mice, stimulation of adipogenesis improve insulin sensitivity because APs and immature adipocytes uptake considerable amounts of glucose and lipids from systemic circulation, thereby reducing the burdens of other insulin target organs muscles. As we showed in the current study, depletion of CD206⁺ cells promotes adipogenesis even in lean mice. That is the reason why CD206⁺ cell depletion improved glucose metabolism even in lean states.

Reviewer's comment 7:

For Fig 8C - this is uninterpretable without some assessment of the efficiency of TGF β ablation with the TAM treatment in these mice.

Our response:

In our revised manuscript, the efficiency of TGF β ablation with the TAM treatment in these mice was described in Supplementary Fig 7b.

b

Reviewer's comment 8:

Fig 9 which is treatment of HFD fed mice with CD206 ablation are uninterpretable without showing the degree to which CD206⁺ are ablated in the model. The data shown in Fig 10a are unclear how they defined the gates that they did. There appear to be CD11c⁺CD206⁺ double positive cells - are those the ones that are actually targeted. The use of fluorescence minus one control would make me feel more comfortable that their gating is justified.

Our response:

In our revised manuscript, we described the gating strategy including the information regarding negative isotype control and used FMO to verify our gating strategy (Supplementary Fig. 12a).

Reviewer's comment 9:

A critical gene to examine in Fig 9-10 is *TGFb1* - is that suppressed to relate to the mechanism of increase in adipogenic regulated genes?

Our response:

We examined the expression of *TGFb1* in eWAT of HFD mice and found that *TGFb1* gene expression was downregulated in DT-treated HFD-fed CD206DTR mice compared to the control littermates (Fig. 9c).

Reviewer #2 (Remarks to the Author):

The present study provides compelling evidence showing that CD206⁺ macrophages in adipose tissues play a role in adipocyte progenitor proliferation and whole-body glucose homeostasis. The authors firstly found that CD206 expression was enriched in M2 macrophages in adipose tissues. Depletion of CD206⁺ cells by DTR stimulated cell proliferation in adipocyte progenitors and also improved glucose homeostasis in vivo. The authors also showed that these actions were mediated through a down-regulation of TGF- β signaling. Overall the data are robust that it clearly suggests the biological importance of CD206⁺ macrophages in adiposity and glucose homeostasis. I would suggest the following points that the authors wish to address to strengthen the authors' conclusions further.

Reviewer's comment 1:

TGF- β in adipose fibrosis has been shown to promote adipose fibrosis. Given the role of CD206⁺ macrophages in TGF- β signaling, the authors should investigate if adipose fibrosis is altered by DTR-depletion of CD206⁺ cells as well as by TGF- β deletion in CD206⁺ cells.

Our response:

We evaluated fibrosis in eWAT by applying the previously reported methods (Watanabe et al, Sci Rep, 2016). Paraffin-embedded tissues were spliced by 4 μ m thickness, placed on slide glasses and subjected to Azan stain. Three images per mice were examined, and fibrotic areas were measured by ImageJ software (National Institutes of Health, MD, USA). As the reviewer pointed, fibrosis was reduced in eWAT by TGF- β deletion in CD206⁺ cells (Supplementary Fig. 8). Although fibrosis in eWAT was not reduced by partial ablation of CD206⁺ cells in CD206DTR mice (Supplementary Fig. 4c), the expressions of some of fibrosis-related genes such as *Col1a1* and *Acta2* were downregulated in CD206⁺ cell-reduced mice (Supplementary Fig. 4b).

Supplementary Figure 4

Supplementary Figure 8.

Reviewer's comment 2:

The role of M2 macrophages in beige adipocyte development has been shown previously. The authors should carefully analyze the browning phenotype in DTR-depletion of CD206⁺ cells. In addition, the data that no difference in body weight seems inconsistent with the role of M2 macrophages in promoting beige fat development; the authors wish to comment on the apparent discrepancy.

Our response:

We assessed the level of browning of iWAT in DT-treated CD206DTR mice. Contrary to the reported role for M2 macrophages in promoting browning, inductions of browning-related genes (Figure 10a) and UCP1 protein (Figure 10b) were significantly up-regulated by partial ablation of CD206⁺ cells. We also found that the percentages of Sca1⁺/PDGFR⁺ were up-regulated in cold-stimulated DT-treated CD206DTR mice (Figure 10c), indicating that the proliferation of beige progenitors (BPs) was stimulated by partial ablation of CD206⁺ cells. Our finding indicates that CD206⁺ cells in iWAT act more as a component of BP niche than a promoter of browning.

Reviewer's comment 3:

Considering the robust changes in systemic glucose tolerance in CD206⁺ cells, it would be insightful to determine the tissues in which glucose uptake is enhanced by CD206-DTR.

Our response:

We performed hyperinsulinemic-euglycemic clamp study to determine the tissue in which glucose uptake was enhanced by CD206-DTR. Statistically significant increments in glucose uptake were detected in GIR and Rd, but not in HGP, in CD206DTR mice (Fig. 9f). Thus, glucose uptake into adipose tissues and muscles was enhanced by partial ablation of CD206 cells.

Reviewer's comment 4:

Concerning the data in Fig. 1h, the authors wish to comment on several M2 markers are disproportionally decreased by CD206-DTR.

Our response:

We observed that about 50% reduction of CD206⁺ cells caused proportional reduction of IL-10 expression (~ 50%) while it disproportionally reduced CD163 and CD209a expressions (> 90%) (Figure 1e). Since IL-10 is known to induce CD163 expression, it may be possible that 50% reduction of IL-10 resulted in much greater reduction of CD163 expression in mice with 50% reduction of M2 macrophages. Although we could not find literatures that report the involvement of IL-10 in CD209a expression, it could also be the case. Thus, we think that M2 markers could disproportionally be decreased depending on gene types.

- 1 Wynn, T. A., Chawla, A. & Pollard, J. W. Macrophage biology in development, homeostasis and disease. *Nature* **496**, 445-455, doi:10.1038/nature12034 (2013).
- 2 Satoh, T. *et al.* Critical role of Trib1 in differentiation of tissue-resident M2-like macrophages. *Nature* **495**, 524-528, doi:10.1038/nature11930 (2013).

Reviewers' Comments:

Reviewer #1:

Remarks to the Author:

The authors have added the required experimental details to assess the paper properly and the clamps are very helpful. A couple of points remain.

1. The use of CD103 as a DC marker is inaccurate as not all DC express CD103 and adipose DC have been shown to have low expression of CD103 protein. The specific marker Zbtb46 would be the most appropriate to be assessed by RT PCR. The high expression of CD11c is more consistent with F4/80+ CD206- cells contaminated with dendritic cells. Overall this is a bit of a minor point = revising the sentence to include the possibility that DC are in the F4/80+ CD206- population is appropriate.

2. In Figure 9a there appears to be a significant dropoff of CD11c+ CD206+ ATMs in the ablation experiment. CD11c+CD206+ cells have been shown to be induced with HFD. Can the authors exclude the possibility that it is the loss of this population that is the main contributor to the improvement in metabolic phenotype? The gates are somewhat arbitrarily chosen to avoid this population.

Reviewer #2:

Remarks to the Author:

The authors addressed the reviewer's comment satisfactorily.

Reviewers' comments:

Response to Reviewer #1:

We are extremely grateful to Reviewer#1 for careful reviewing of our previous manuscript. Thanks to the Reviewer's valuable suggestions, we believe that the quality of our revised manuscript have been greatly up-graded.

Reviewer #1 (Remarks to the Author):

The authors have added the required experimental details to assess the paper properly and the clamps are very helpful. A couple of points remain.

1. The use of CD103 as a DC marker is inaccurate as not all DC express CD103 and adipose DC have been shown to have low expression of CD103 protein. The specific marker *Zbtb46* would be the most appropriate to be assessed by RT PCR. The high expression of CD11c is more consistent with F4/80+ CD206- cells contaminated with dendritic cells. Overall this is a bit of a minor point = revising the sentence to include the possibility that DC are in the F4/80+ CD206- population is appropriate.

Response;

According the reviewer's comment, we examined *Zbtb46* expressions in F4/80⁺CD206⁻ and F4/80⁺CD206⁺ fractions. As shown in Fig. 1b in our revised manuscript, *Zbtb46* was predominantly expressed in F4/80⁺CD206⁻ ATMs, whereas F4/80⁺CD206⁺ ATMs expressed only a negligible level of *Zbtb46*. These finding support our claim that DCs belong to F4/80⁺CD206⁻ fraction rather than F4/80⁺CD206⁺ fraction. In the main text in our revised manuscript, we added a description regarding this point (page 6, line 2).

2. In Figure 9a there appears to be a significant dropoff of CD11c+ CD206+ ATMs in the ablation experiment. CD11c+CD206+ cells have been shown to be induced with HFD. Can the authors exlude the possibility that it is the loss of this population that is the main contributor to the improvement in metabolic phenotype? The gates are somewhat arbitrarily chosen to avoid this population.

Response;

According the reviewer's comment, we re-analyzed the flow cytometry data by carefully re-setting the gating procedure. By applying a new gating method, which was designed by an

immunologist who has expert skills on flow cytometry, we found that CD206+CD11c+ ATMs indeed existed in eWAT of HFD mice as the reviewer pointed. As shown in Fig. 9a in our revised manuscript, DT-treatment depleted not only CD206+CD11c- but also CD206+CD11c+ ATMs. According the suggestion by the Reviewer, we added the following sentence in the discussion section of the main text (page 15, line 16): **In HFD-fed mice, both CD11c+CD206- and CD11c+CD206+ ATMs are markedly increased; DT-treatment depleted not only CD206+CD11c- but also CD206+CD11c+ ATMs. Although it seems that CD206+ M2-like ATMs plays a major role in the insulin-sensitive phenotype-associated inflammation, the possibility that depletion of CD206+CD11c+ ATMs contributes to this phenotype may not completely be excluded.**